# Structural basis for PRC2 decoding of active histone methylation marks H3K36me2/3

**Ksenia Finogenova[1], Jacques Bonnet[1], Simon Poepsel[2,3,4], Ingmar B Schäfer[5], Katja Finkl[1], Katharina Schmid[1], Claudia Litz[1], Mike Strauss[6†], Christian Benda[5], Jürg Müller[1]***

[1]Max Planck Institute of Biochemistry, Laboratory of Chromatin Biology, Martinsried, Germany; [2]California Institute for Quantitative Biology (QB3), University of California, California Institute for Quantitative Biology (QB3), Molecular Biophysics and Integrative Bio-Imaging Division, Lawrence Berkeley National Laboratory, Berkeley, United States; [3]University of Cologne, Center for Molecular Medicine Cologne (CMMC), Faculty of Medicine and University Hospital Cologne, Cologne, Germany; [4]Cologne Excellence Cluster for Cellular Stress Responses in Ageing-Associated Diseases (CECAD), University of Cologne, Cologne, Germany; [5]Max Planck Institute of Biochemistry, Department of Structural Cell Biology, Martinsried, Germany; [6]Max Planck Institute of Biochemistry, cryoEM Facility, Martinsried, Germany

***For correspondence:**
muellerj@biochem.mpg.de

**Present address:** [†]McGill University, Department of Anatomy and Cell Biology, Montreal, Canada

**Competing interests:** The authors declare that no competing interests exist.

**Abstract** Repression of genes by Polycomb requires that PRC2 modifies their chromatin by trimethylating lysine 27 on histone H3 (H3K27me3). At transcriptionally active genes, di- and trimethylated H3K36 inhibit PRC2. Here, the cryo-EM structure of PRC2 on dinucleosomes reveals how binding of its catalytic subunit EZH2 to nucleosomal DNA orients the H3 N-terminus via an extended network of interactions to place H3K27 into the active site. Unmodified H3K36 occupies a critical position in the EZH2-DNA interface. Mutation of H3K36 to arginine or alanine inhibits H3K27 methylation by PRC2 on nucleosomes *in vitro*. Accordingly, *Drosophila* H3K36A and H3K36R mutants show reduced levels of H3K27me3 and defective Polycomb repression of HOX genes. The relay of interactions between EZH2, the nucleosomal DNA and the H3 N-terminus therefore creates the geometry that permits allosteric inhibition of PRC2 by methylated H3K36 in transcriptionally active chromatin.

## Introduction

Many post-translational modifications on histone proteins are essential for processes in the underlying chromatin. Typically, histone modifications themselves do not alter chromatin structure directly but function by binding effector proteins which alter chromatin or by interfering with such interactions. The histone methyltransferase Polycomb Repressive Complex 2 (PRC2) and its regulation by accessory proteins and histone modifications represent a prime example for understanding these interaction mechanisms (*Laugesen et al., 2019*; *Yu et al., 2019*).

PRC2 trimethylates lysine 27 in histone H3 (H3K27me3), a modification that is essential for the transcriptional repression of developmental regulator genes that control cell fate decisions in metazoans (*Pengelly et al., 2013*; *McKay et al., 2015*). H3K27me3 marks chromatin for interaction with PRC1 (*Fischle et al., 2003*; *Min et al., 2003*), an effector which compacts chromatin (*Francis et al., 2004*; *Grau et al., 2011*). H3K27me3 is also recognized by PRC2 itself, and this interaction

allosterically activates the PRC2 enzyme complex to facilitate deposition of H3K27me3 across extended domains of chromatin (*Hansen et al., 2008*; *Margueron et al., 2009*; *Jiao and Liu, 2015*).

Genetic studies and subsequent biochemical work established that PRC2 is in addition subject to negative regulation. In particular, the H3K4me3, H3K36me2, and H3K36me3 marks present on nucleosomes in transcriptionally active chromatin directly inhibit H3K27 methylation by PRC2 (*Klymenko and Müller, 2004*; *Schmitges et al., 2011*; *Yuan et al., 2011*; *Gaydos et al., 2012*; *Streubel et al., 2018*). Importantly, while stimulation of PRC2 activity by H3K27me3 acts in *trans*, inhibition of PRC2 by H3K4me3, H3K36me2, and H3K36me3 requires that these modifications are present in *cis*, that is, on the same H3 molecule containing the K27 substrate lysine (*Schmitges et al., 2011*; *Yuan et al., 2011*; *Voigt et al., 2012*). While recent structural studies have uncovered the allosteric activation mechanism for PRC2 (*Jiao and Liu, 2015*; *Justin et al., 2016*), the molecular basis of PRC2 inhibition by active chromatin marks has remained enigmatic. In particular, in nucleosome-binding assays, PRC2–DNA interactions make the largest contribution to the nucleosome-binding affinity of PRC2 (*Wang et al., 2017*; *Choi et al., 2017*) and H3K4me3, H3K36me2 and H3K36me3 do not seem to have a major effect on this binding affinity (*Schmitges et al., 2011*; *Guidotti et al., 2019*; *Jani et al., 2019*). Instead, these three modifications were found to reduce the $k_{cat}$ of PRC2 for H3K27 methylation (*Schmitges et al., 2011*; *Jani et al., 2019*). Recent cross-linking studies led to the suggestion of a possible sensing pocket for H3K36 on the surface of EZH2 (*Jani et al., 2019*) but there is no structural data how this proposed interaction might occur. Similarly, a recent structure of PRC2 bound to a dinucleosome revealed how the catalytic lobe of PRC2 contacts nucleosomes through DNA interactions but provided no structural insight into how the H3 N-termini might be recognized (*Poepsel et al., 2018*).

Here, a refined structure of PRC2 bound to a dinucleosome allowed us to visualize how the histone H3 N-terminus on substrate nucleosomes is threaded into the EZH2 active site. Our analyses reveal that H3K36 assumes a critical position in the PRC2-nucleosome interaction interface that permits the complex to gauge the H3K36 methylation state.

## Results

### EZH2 interaction with nucleosomal DNA orients the H3 N-terminus for H3K27 binding to the active site

We assembled recombinant full-length human PRC2 in complex with its accessory factor PHF1 (i.e. PHF1-PRC2) (*Choi et al., 2017*) on a heterodimeric dinucleosome (di-Nuc), which consisted of a 'substrate' nucleosome with unmodified histone H3 and an 'allosteric' nucleosome containing H3 with a trimethyllysine analog (*Simon et al., 2007*) at K27, separated by a 35 base pair (bp) DNA linker (*Poepsel et al., 2018*; *Figure 1A,B*). Single-particle cryo-electron microscopy analysis yielded a reconstruction of the PHF1-PRC2:di-Nuc assembly with an overall resolution of 5.2 Å (*Figure 1—figure supplement 1*, *Figure 1—figure supplement 2*, *Figure 1—figure supplement 3*). The map showed clear density for the catalytic lobe of PRC2 with similar chromatin interactions and binding geometry as previously described for the catalytic lobe of AEBP2-PRC2 (*Poepsel et al., 2018*) where PRC2 contacts the two nucleosomes via interactions with the DNA gyres (*Figure 1C*). Specifically, the substrate nucleosome is bound by the EZH2$_{CXC}$ domain residues K563, Q565, K569 and Q570 (*Figure 1D*, *Figure 1—figure supplement 4A*, cf. *Poepsel et al., 2018*), while the allosteric nucleosome is contacted by EED and by the SBD and SANT1 domains of EZH2 (*Figure 1E*, *Poepsel et al., 2018*). We could not detect density for the 'bottom lobe' of PRC2 (*Chen et al., 2018*; *Kasinath et al., 2018*) or for the N-terminal winged-helix and tudor domains of PHF1 that bind DNA and H3K36me3, respectively (*Choi et al., 2017*; *Li et al., 2017*; *Ballaré et al., 2012*; *Cai et al., 2013*; *Musselman et al., 2013*).

Using particle signal subtraction and focused refinement on the interface of EZH2 and the substrate nucleosome (*Figure 1—figure supplement 2*, *Figure 1—figure supplement 3*), we then obtained an improved map at an apparent overall resolution of 4.4 Å which revealed well-defined density for the H3 N-terminus (*Figure 1F*, *Figure 1—figure supplement 3B–D*). The visible side-chain density combined with the crystallographic models of the PRC2 catalytic lobe and of the mononucleosome enabled us to build a pseudo-atomic model of the histone H3 N-terminus spanning residues R26 to K37 (*Figure 1F*). This model revealed that EZH2 recognizes the H3 N-terminus

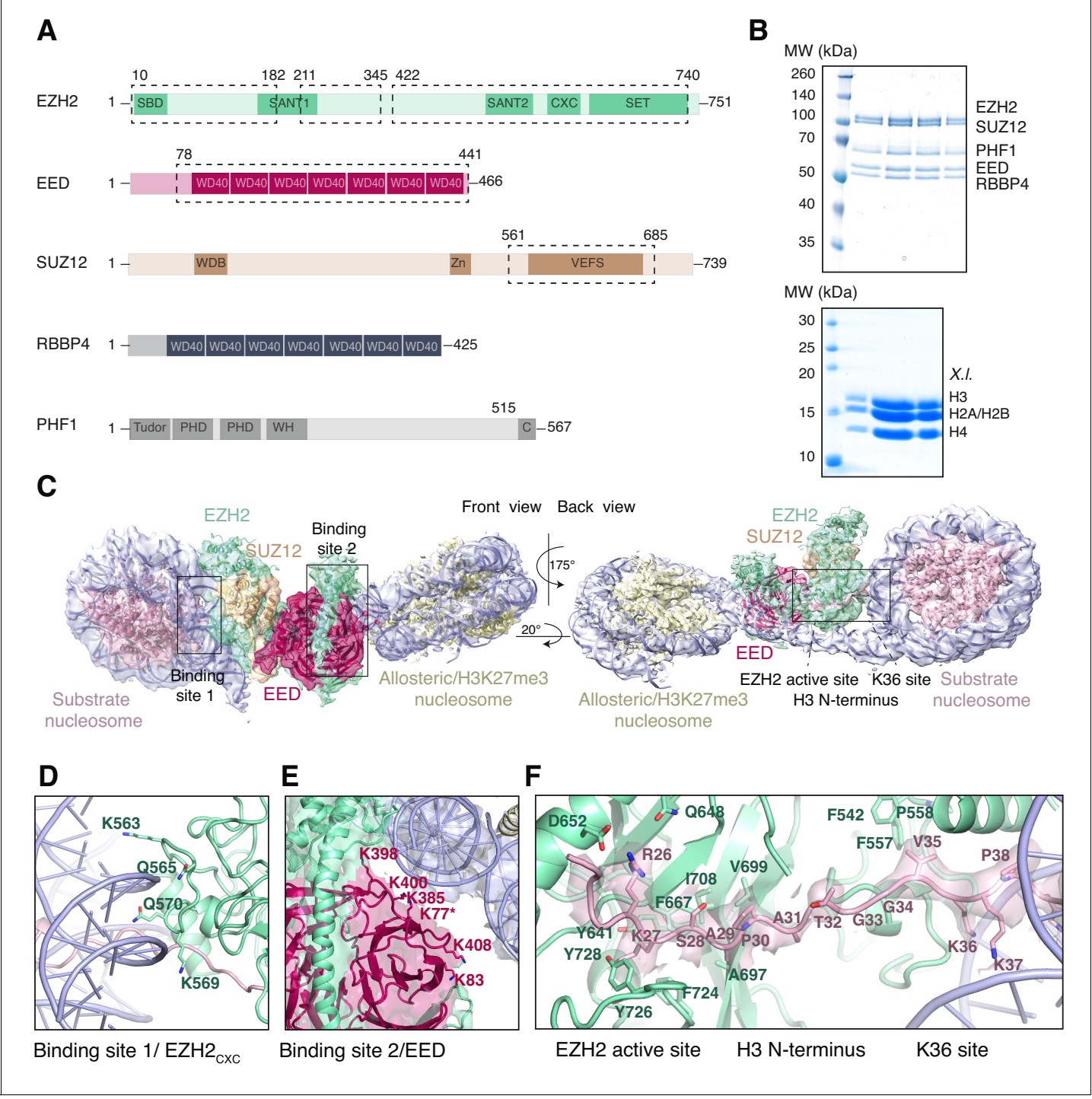

**Figure 1.** Interaction of the PRC2 catalytic lobe with nucleosomal DNA orients the H3 N-terminus for H3K27 binding to the active site. (A) Domain organization in the five subunits of PHF1-PRC2. Dashed boxes indicate protein portions visible in the PHF1-PRC2:di-Nuc cryo-EM reconstruction and fitted in the structural model. In PHF1, C corresponds to the short C-terminal fragment used in PHF1_C-PRC2. (B) Coomassie-stained SDS PAGE analysis of representative PHF1-PRC2 (upper panel) and *Xenopus laevis* (*X.l.*) octamer preparations (lower panel) after size-exclusion chromatography (SEC) purification. Pooled fractions of PHF1-PRC2, incubated with heterodimeric dinucleosomes generated by DNA ligation of a reconstituted unmodified and a H3Kc27me3-modified mononucleosome as described in *Poepsel et al., 2018* were used as input material for cryo-EM analysis. (C) Cryo-EM reconstruction of PHF1-PRC2:di-Nuc in two orientations with fitted crystal structures of human PRC2 catalytic lobe (PDB: 5HYN, *Justin et al., 2016*) and nucleosomes (1AOI, *Luger et al., 1997*) in a di-Nuc model with 35 bp linker DNA (see also *Figure 1—figure supplements 1–4*, *Supplementary file 1*, *Figure 1—video 1*, *Source code 1*). Density is colored as in (A) to show PRC2 subunits, DNA (blue) and octamers of substrate (pink) and allosteric

*Figure 1 continued on next page*

*Figure 1 continued*

(yellow) nucleosomes. Boxes indicate regions shown in (D), (E) and (F), respectively. (D) Interaction of EZH2$_{CXC}$ residues with the DNA gyres of the substrate nucleosome; residues mutated in PRC2$^{CXC>A}$ are indicated. For the H3 N-terminus (pink), only the peptide backbone is shown in this view (see F). (E) Interface formed by EED and the EZH2 SBD domain with DNA gyres on the allosteric nucleosome; residues mutated in PRC2$^{EED>A}$ are indicated. Asterisk indicates the approximate location of a residue, which is not built in the model. (F) The H3 N-terminus (pink), shown as a pseudoatomic model fitted into the 4.4 Å density map, is recognized by EZH2 through an extensive interaction network (see text). Note the well-defined side-chain density of H3K36 (see also *Figure 1—figure supplement 3D* and *Figure 1—figure supplement 4C–E*).

The online version of this article includes the following video and figure supplement(s) for figure 1:

**Figure supplement 1.** Initial Cryo-EM analysis of the PHF1-PRC2:di-Nuc complex (related to *Figure 1*).
**Figure supplement 2.** Overview of the cryo-EM Data-Processing and Particle Sorting Scheme (related to *Figure 1*).
**Figure supplement 3.** Cryo-EM analysis of the focused EZH2$_{sub}$-Nuc$_{sub}$ map (related to *Figure 1*).
**Figure supplement 4.** The improved map of the interaction between EZH2 and the substrate nucleosome after focused refinement reveals location of H3K36 and its environment (related to *Figure 1*).
**Figure 1—video 1.** Cryo-EM structure of the PHF1-PRC2:di-Nuc complex (related to *Figure 1*).
https://elifesciences.org/articles/61964#fig1video1

via an extended network of contacts besides the previously described ionic interactions near the active site where H3 R26 interacts with EZH2 Q648/D652, and H3 K27 with the aromatic cage above the EZH2 catalytic center (*Justin et al., 2016*; *Figure 1F*). Specifically, our structure suggests two hydrophobic hotspots, the first one involving H3 A29/P30 and EZH2 residues F667, A697, V699, I708 and F724 and the second one involving H3 V35 and F542, F557 and P558 of EZH2 (*Figure 1F*). H3 G33/G34 is likely not recognized by PRC2 but might act as a flexible hinge between the two hydrophobic interaction sites (*Figure 1F*). H3K36 is directly juxtaposed to the EZH2$_{CXC}$-DNA interaction surface and appears to be involved in the EZH2-DNA interface. The side-chain density of H3K36 suggests that the epsilon-amino group of H3K36 engages in a polar interaction with the carbonyl group of Q570 and possibly in long-range electrostatic interactions with the phosphate backbone of the nucleosomal DNA (*Figure 1F*, *Figure 1—figure supplement 4C–E*). Taken together, our analyses reveal an extensive network of interactions between EZH2, the nucleosomal DNA and the H3 N-terminus. This complex geometric arrangement orients the H3 N-terminus into an extended conformation, threading H3K27 into the EZH2 active site. In this context, it should be noted that a previously postulated H3K36-binding pocket centered on E579 of EZH2 (*Jani et al., 2019*) is located approximately 19 Å away from H3K36 in our structure (*Figure 1—figure supplement 4F*). An interaction of H3K36 with E579 of EZH2 as proposed by Muir and co-workers (*Jani et al., 2019*) would require a very different binding geometry of PRC2 on the nucleosome and major structural rearrangements of PRC2 or the nucleosome in order to avoid steric clashes.

## The EZH2 CXC contact with DNA is essential for H3K27 methylation

We next analyzed how the PRC2 surfaces contacting the substrate and the allosteric nucleosome contribute to the formation of productive PRC2-chromatin interactions. For these experiments, we used PHF1$_C$-PRC2, which contains the minimal 5 kDa PRC2-interaction domain of PHF1 (*Figure 1A*, *Choi et al., 2017*; *Chen et al., 2020*) but lacks the H3K36me3-binding tudor and the DNA-binding winged-helix domains of PHF1 (*Musselman et al., 2013*; *Choi et al., 2017*; *Li et al., 2017*). PHF1$_C$-PRC2 therefore only retains the DNA-binding surfaces of the 4-subunit PRC2 core complex and was used because it generally behaved better in purifications than the 4-subunit PRC2 core complex. For simplicity we shall, in the following, refer to the PHF1$_C$-PRC2 complex as PRC2. We generated three mutant versions of PRC2. In PRC2$^{CXC>A}$ (K563A Q565A K569A Q570A), the EZH2$_{CXC}$ interface is mutated (*Figure 1D*), in PRC2$^{EED>A}$ (K77A K83A K385A K398A K400A K408A), the EED interface contacting the allosteric nucleosome (*Figure 1E*), is mutated, and PRC2$^{CXC>A/EED>A}$ carries the combination of these mutations. We first used electromobility shift assays (EMSA) to measure the binding affinity of wild-type and mutant PRC2 complexes on mononucleosomes. These mononucleosomes were assembled on a 215 bp long DNA fragment containing the 147 bp 601 nucleosome-positioning sequence (*Lowary and Widom, 1998*) in the center and linker DNA on both sides. Wild-type PRC2 bound this mononucleosome with an apparent K$_d$ in the mid-nanomolar range (*Figure 2A,B*, cf. *Choi et al., 2017*). The binding affinities of PRC2$^{CXC>A}$ or PRC2$^{EED>A}$ were two- to three-fold lower than that of wild-type PRC2 and that of PRC2$^{CXC>A/EED>A}$ was about five-

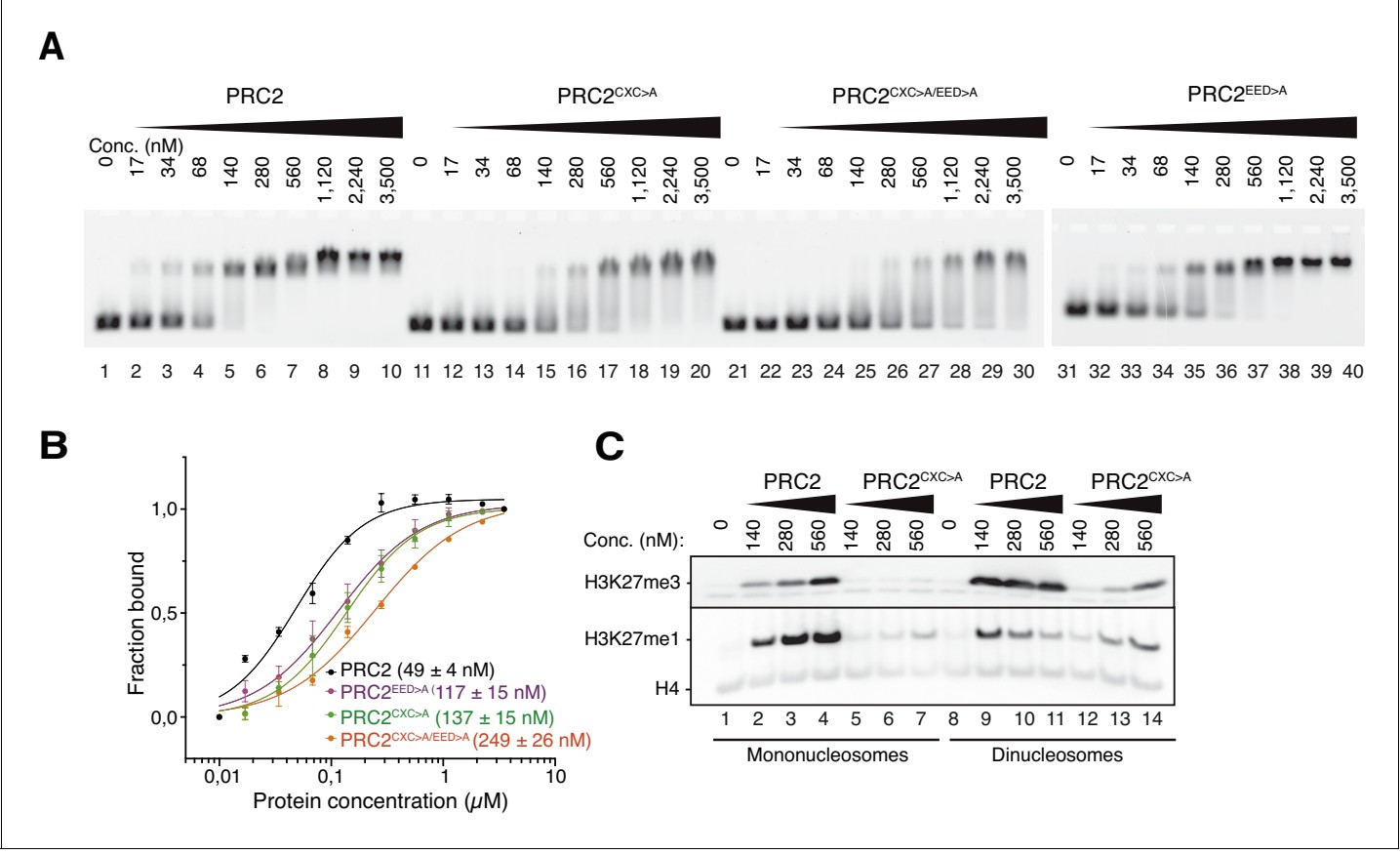

**Figure 2.** The EZH2$_{CXC}$-DNA interaction interface is critical for H3K27 methylation on nucleosomes. (**A**) Binding reactions with indicated concentrations of PRC2 (lanes 1–10), PRC2$^{CXC>A}$ (lanes 11–20), PRC2$^{CXC>A/EED>A}$ (lanes 21–30), or PRC2$^{EED>A}$ (lanes 31–40) and 45 nM 6-carboxyfluorescein-labeled mononucleosomes, analyzed by EMSA on 1.2% agarose gels. The EMSA with PRC2$^{EED>A}$ was run on a separate gel. (**B**) Quantitative analysis of EMSA data in A by densitometry of 6-carboxyfluorescein signals from independent experiments (n = 3); error bars, SEM. (**C**) Western Blot (WB) analysis of H3K27me1 and H3K27me3 formation in HMTase reactions with indicated concentrations of PRC2 and PRC2$^{CXC>A}$ on 446 nM mononucleosomes (lanes 1–7) or 223 nM dinucleosomes (lanes 8–14). Note that these concentrations result in equal numbers of nucleosomes and therefore equal numbers of H3 substrate molecules in the reactions on mono- and dinucleosomes, as can be seen from the Coomassie-stained gel of the reactions in **Figure 2—figure supplement 1B**. H4 WB signal served as control for western blot processing.

The online version of this article includes the following figure supplement(s) for figure 2:

**Figure supplement 1.** The EZH2$_{CXC}$-DNA interaction interface is critical for H3K27 methylation on nucleosomes (related to **Figure 2**).

fold lower compared to the wild-type complex (**Figure 2A,B**, compare lanes 11–40 with 1–10). The PRC2$^{CXC>A/EED>A}$ complex therefore still binds to nucleosomes with sub-micromolar affinity (**Figure 2A**, lanes 21–30). Nucleosome binding by the PRC2$^{CXC>A/EED>A}$ complex could in part be due to incomplete disruption of the mutated interfaces but it likely also represents nucleosome binding contributed by the bottom lobe of PRC2 comprising the N-term of SUZ12 and RBBP4 (**Chen et al., 2018**; **Nekrasov et al., 2005**). In particular, biochemical studies on *Drosophila* PRC2 originally found that a minimal complex formed between Su(z)12 and the RBBP4 ortholog Caf1-55 binds to nucleosomes (**Nekrasov et al., 2005**). Moreover, negative stain EM analyses of human PRC2 bound to a dinucleosome identified several 2D classes where the bottom lobe contacts one or two of the two nucleosomes (**Poepsel et al., 2018**).

The binding affinity of the PRC2 core complex for chromatin therefore appears to result from interactions of at least three distinct complex surfaces with nucleosomes, the EZH2$_{CXC}$ domain, the EED nucleosome-binding interface and the SUZ12$_N$:RBBP4 lobe. Considering the architecture of the catalytic lobe (**Figure 1C**) and of the isolated full PRC2 core complex (**Kasinath et al., 2018**), it is very unlikely that the EZH2$_{CXC}$ domain and the EED nucleosome-binding interface could simultaneously engage with the same nucleosome at a time. Finally, we note that in EMSAs monitoring

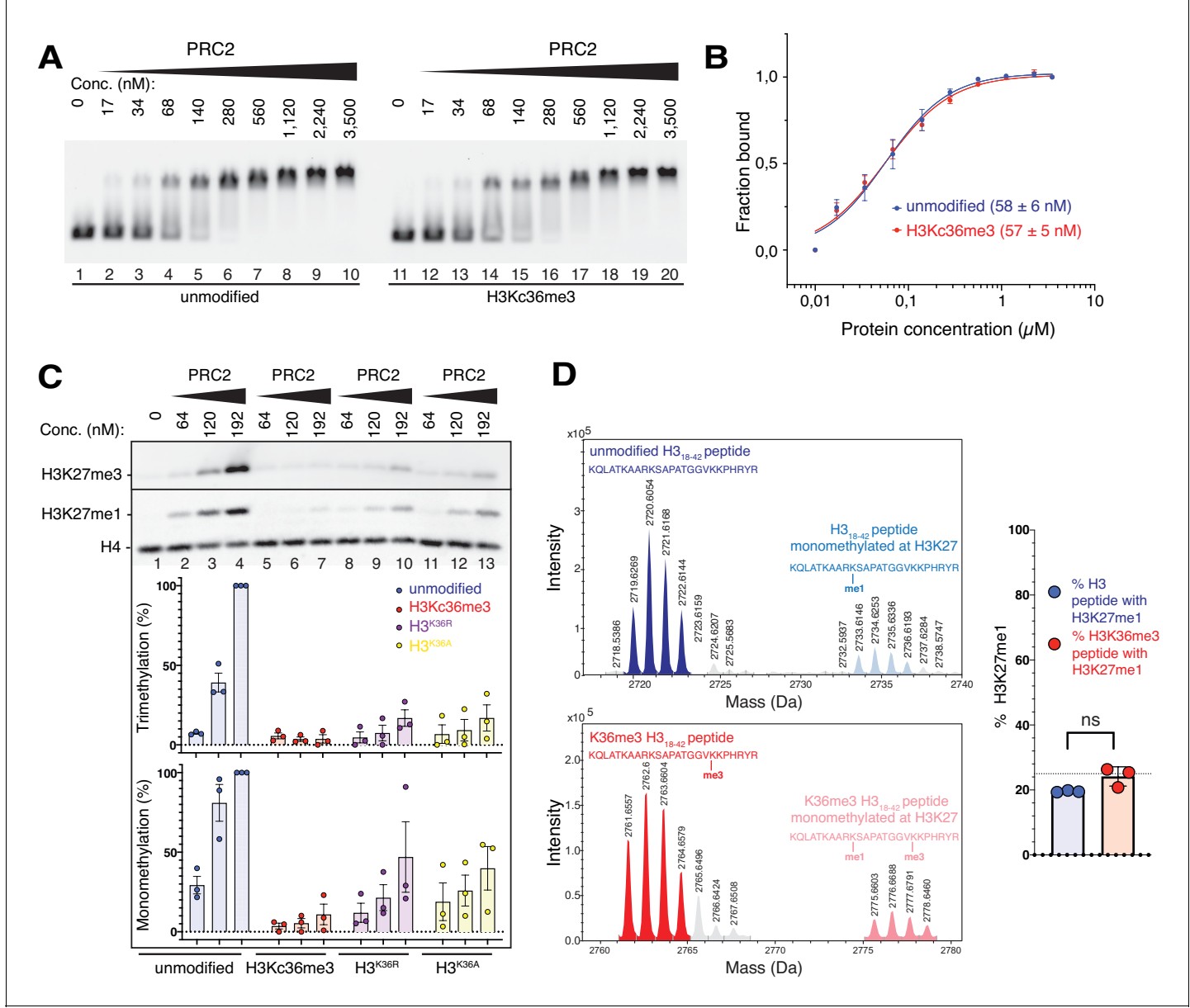

**Figure 3.** The unmodified H3K36 side chain in the EZH2$_{CXC}$-DNA interaction interface is critical for H3K27 methylation on nucleosomes. (A, B) EMSA analysis and quantification as in *Figure 2A and B*, using PRC2 and mononucleosomes that were unmodified (lanes 1–10) or contained a trimethyllysine analog at H3K36 (H3Kc36me3, lanes 11–20) (*Simon et al., 2007*). (C) Western blot (WB) analysis of HMTase reactions with PRC2 as in *Figure 2C* on unmodified (lanes 1–4), H3Kc36me3 (lanes 5–7), H3$^{K36R}$ (lanes 8–10) or H3$^{K36A}$ (lanes 11–13) mononucleosomes (446 nM). Coomassie-stained gel of reactions is shown in *Figure 3—figure supplement 1A*. Bottom: quantification of H3K27me3 and H3K27me1 chemiluminescence signals, respectively, by densiometry analysis from three independent experiments. In each experiment, the methylation signal in lane four was defined as 100% and used to quantify the corresponding H3K27 methylation signals in the other lanes on the same membrane. Circles show individual data points and error bars SEM. (D) HMTase reactions monitoring H3K27me1 formation by PRC2 on H3$_{18-42}$ peptides that were unmodified (top) or contained K36me3 (bottom). Left: Deconvoluted ESI-MS spectra from data shown in *Figure 3—figure supplement 1B*. On both substrates, areas of the four colored peaks of H3K27me1-modified and unmodified substrate peptides were used for quantification of H3K27me1 formation. Right: Symbols represent percentages of peptides carrying H3K27me1 in technical triplicate experiments, error bars show SD; Welch's t-test showed no significant (ns) difference between H3K27 monomethylation on the two peptide substrates.

The online version of this article includes the following figure supplement(s) for figure 3:

**Figure supplement 1.** Accommodation of unmodified H3K36 in the EZH2$_{CXC}$-DNA interaction interface is essential for H3K27 methylation on nucleosomes and PHF1-PRC2 (related to *Figure 3*).

binding of PRC2 to a dinucleosome, we observed a complex mixture of slowly migrating species and this has precluded experiments aimed at discriminating between binding events involving specific PRC2 surfaces with dinucleosomes. In conclusion, the structural data (*Figure 1C,D* and *Poepsel et al., 2018*) suggest that a key interaction of PRC2 with substrate nucleosomes occurs through contacts of the EZH2$_{CXC}$ domain with the DNA gyres, whereas the biochemical analyses argue that this interaction contributes only modestly to the overall chromatin-binding affinity of the complex.

We next analyzed how mutation of the DNA-contacting residues in the EZH2$_{CXC}$ domain affects H3K27 methylation by PRC2. On the same mononucleosomes used above, PRC2$^{CXC>A}$ showed almost no detectable histone methyltransferase (HMTase) activity compared to wild-type PRC2 (*Figure 2C*, compare lanes 5–7 with 2–4, see also *Figure 2—figure supplement 1A*). On dinucleosomes, EED binding to one nucleosome might be expected to enable interaction of the mutated EZH2$^{CXC>A}$ domain with the H3 N-termini on the juxtaposed second nucleosome and thereby - at least partially - restore H3K27 methylation. Indeed, on dinucleosomes, the PRC2$^{CXC>A}$ complex does generate H3K27me1 and -me3 although still much less efficiently than wild-type PRC2 (*Figure 2C*, compare lanes 12–14 with 9–11). When comparing the activity of wild-type PRC2 and the PRC2$^{CXC>A}$ complex, it is important to keep in mind that on dinucleosomes interpretation of H3K27me1 and -me3 formation as read-out for complex activity is considerably more complicated than on mononucleosomes because H3K27me3, once placed on one of the nucleosomes, will allosterically activate PRC2 to methylate H3K27 on the linked second nucleosome (*Margueron et al., 2009*; *Jiao and Liu, 2015*).

To complement these experiments, we also compared the HMTase activity of wild-type PRC2 and PRC2$^{CXC>A}$ complex on free histone H3$_{18-42}$ peptides using a mass spectrometry-based assay to detect H3K27 methylation. It is well established that wild-type PRC2 methylates K27 on free histone H3 with much lower efficiency than on H3 in nucleosomes (*Cao et al., 2002*; *Czermin et al., 2002*; *Kuzmichev et al., 2002*; *Müller et al., 2002*), and this is also recapitulated in our assays on H3$_{18-42}$ peptides where we primarily detect H3K27me1 but no H3K27me3 formation, even after extended incubation of the reaction (see *Figure 2—figure supplement 1B* and compare with *Figure 2C*). However, it is important to note that PRC2$^{CXC>A}$ did not show reduced K27 methylation activity compared to wild-type PRC2 on this H3 peptide substrate (*Figure 2—figure supplement 1B*). The mutations in the EZH2$^{CXC}$ domain therefore do not appear to alter the conformation of EZH2 in a way that would directly interfere with catalysis. Taken together, these observations strongly argue for a mechanism where interaction of the EZH2$_{CXC}$ domain with the DNA on the substrate nucleosomes is a critical step for engaging the H3 N-terminus in a manner that allows effective H3K27 methylation.

## Unmodified H3K36 in the EZH2$_{CXC}$-DNA interaction interface is critical for H3K27 methylation in nucleosomes

The placement of H3K36 in the EZH2$_{CXC}$-DNA interface (*Figure 1F*) suggested that even though a tri- or di-methylated K36 side chain could theoretically be accommodated in this interface, these modified side chains might provide a less optimal fit and thereby inhibit H3K27 methylation. In EMSAs, the affinity of PRC2 for binding to mononucleosomes containing a trimethyllysine analog at H3K36 (H3Kc36me3) (*Simon et al., 2007*) was indistinguishable from that for binding to unmodified mononucleosomes (*Figure 3A,B*). As previously reported (*Schmitges et al., 2011*; *Yuan et al., 2011*), on H3Kc36me3-containing mononucleosomes, H3K27 mono- and trimethylation by PRC2 was more than 10-fold inhibited (*Figure 3C*, compare lanes 5–7 with 2–4, see also *Figure 3—figure supplement 1A*). Methylation of H3K27 was also inhibited on mononucleosomes where H3K36 had been mutated to arginine (H3$^{K36R}$) and, intriguingly, also on H3$^{K36A}$ mononucleosomes (*Figure 3C*, compare lanes 8–13 with 2–4). PRC2 inhibition on H3$^{K36R}$ and H3$^{K36A}$ mononucleosomes was less severe than on H3Kc36me3 mononucleosomes (*Figure 3C*, compare lanes 8–13 with 5–7). We note that the quantitative analyses here show inhibition of PRC2 HMTase activity on H3$^{K36A}$ mononucleosomes, consistent with earlier studies (*Jani et al., 2019*), whereas other studies previously had failed to detect inhibition on H3$^{K36A}$ mononucleosomes (*Schmitges et al., 2011*). Taken together, productive positioning of H3K27 in the catalytic center of PRC2 appears to be exquisitely sensitive to the chemical nature of the H3K36 side chain. Neither the side chains of trimethyllysine or arginine nor

the short apolar side chain of alanine appear to provide the correct fit at the position of residue 36 in H3.

To extend these analyses, we compared PRC2 HMTase activity on histone $H3_{18-42}$ peptides that were either unmodified or contained H3K36me3. Importantly, on this free H3 peptide, H3K36me3 did not inhibit H3K27 monomethylation by PRC2 (*Figure 3D*, *Figure 3—figure supplement 1B*). This is consistent with previous studies reporting that on peptide substrates H3K36me3 only has a minor effect on the $k_{cat}$ of H3K27 methylation (*Schmitges et al., 2011*; *Jani et al., 2019*). The allosteric inhibition of PRC2 by H3K36me3 therefore only occurs in the context of the geometric constraints of the nucleosome.

## H3K36me3 inhibits H3K27 methylation by PHF1-PRC2

DNA-binding by the winged-helix domain of PHF1 increases the binding affinity and residence time of PHF1-PRC2 on nucleosomes about two- to three-fold, resulting in more efficient H3K27 methylation by this complex compared to PRC2 (*Choi et al., 2017*). Furthermore, the PHF1 tudor domain binds to H3K36me3 in the context of a nucleosome (*Musselman et al., 2013*) and this interaction has been reported to inhibit PHF1-PRC2 from tri-methylating H3K27 on H3K36me3-containing chromatin isolated from yeast cells (*Musselman et al., 2012*). To analyze how H3K36me3 inhibits PHF1-PRC2 in our fully recombinant system, we compared the HMTase activity of full-length PHF1-PRC2 (*Figure 1B*) on unmodified and H3Kc36me3 mononucleosomes. H3K27 mono- and tri-methylation by PHF1-PRC2 was strongly inhibited on H3Kc36me3 mononucleosomes (*Figure 3—figure supplement 1C*). H3K36me3 therefore inhibits H3K27 methylation by PHF1-PRC2 even though this complex has higher binding affinity and a prolonged residence time on nucleosomes (*Choi et al., 2017*). In Polycomblike, the *Drosophila* ortholog of PHF1, a region comprising the tudor domain and the adjacent PHD finger has been reported to bind H3K36me3, H3K4me3, H3K9me3 and, more weakly, also H3K14me3 and H3K27me3 (*Ballaré et al., 2012*). We note, however, that the tudor domain of Polycomblike contains an incomplete aromatic cage and, on its own, is unable to bind methylated lysines (*Friberg et al., 2010*). Further analyses will be needed to assess whether and how interaction of PHF1 or Polycomblike with H3K36me3 might change H3K27 methylation by PRC2 on more complex oligonucleosome substrates that contain both H3K36me3-modified and unmodified nucleosomes.

## *Drosophila* with H3$^{K36R}$ or H3$^{K36A}$ mutant chromatin arrest development after completion of embryogenesis

The observation that PRC2 is not only inhibited on H3K36me2/3-modified nucleosomes but also on H3$^{K36R}$ and on H3$^{K36A}$ mutant nucleosomes prompted us to investigate how H3K27 trimethylation is affected in *Drosophila* with H3$^{K36R}$ or H3$^{K36A}$ mutant chromatin. H3K27me3 is primarily found on canonical histone H3 (*Pengelly et al., 2013*; *McKay et al., 2015*). We used the following strategy to replace the canonical histone H3 gene copies encoded in the *HisC* gene cluster with H3$^{K36R}$ or H3$^{K36A}$ mutant versions. Animals that are homozygous for a deletion of the *HisC* gene cluster (i.e. *Df(2L)HisC* homozygotes) arrest development at the blastoderm stage after exhaustion of the pool of maternally-deposited histones but transgene cassettes providing 12 copies of the wild-type histone gene unit (*12xHisGU$^{WT}$*) rescue *Df(2L)HisC* homozygotes into viable adults (*McKay et al., 2015*; *Günesdogan et al., 2010*). We therefore generated *Df(2L)HisC* homozygotes carrying *12xHisGU$^{H3K36R}$* or *12xHisGU$^{H3K36A}$* transgene cassettes and shall refer to these animals as *H3$^{K36R}$* and *H3$^{K36A}$* mutants, respectively. For the analysis of *H3$^{K36R}$* mutants, we used a strain generated by Matera and colleagues that carried a single *12xHisGU$^{H3K36R}$* array (*McKay et al., 2015*). We used the *12xHisGU* transgene strategy developed by Herzig and colleagues (*Günesdogan et al., 2010*) that relies on the use of multiple copies of *3xHisGU* miniarrays to build strains that allowed us to generate *H3$^{K36A}$* and, as additional control, also *H3$^{K36R}$* mutant animals.

Using the *H3$^{K36R}$* strain from Matera and colleagues (*McKay et al., 2015*), we found that *H3$^{K36R}$* mutant animals complete embryogenesis and that their cuticle morphology is indistinguishable from *wildtype* (*Figure 4*). In agreement with the results from Matera and colleagues (*McKay et al., 2015*), we found that these animals arrest development during the larval or pupal stages. Specifically, 81% of *H3$^{K36R}$* mutant animals arrested development at variable time points during larval growth, 18% developed to the end of third larval instar and formed pupae that died prior to metamorphosis, and

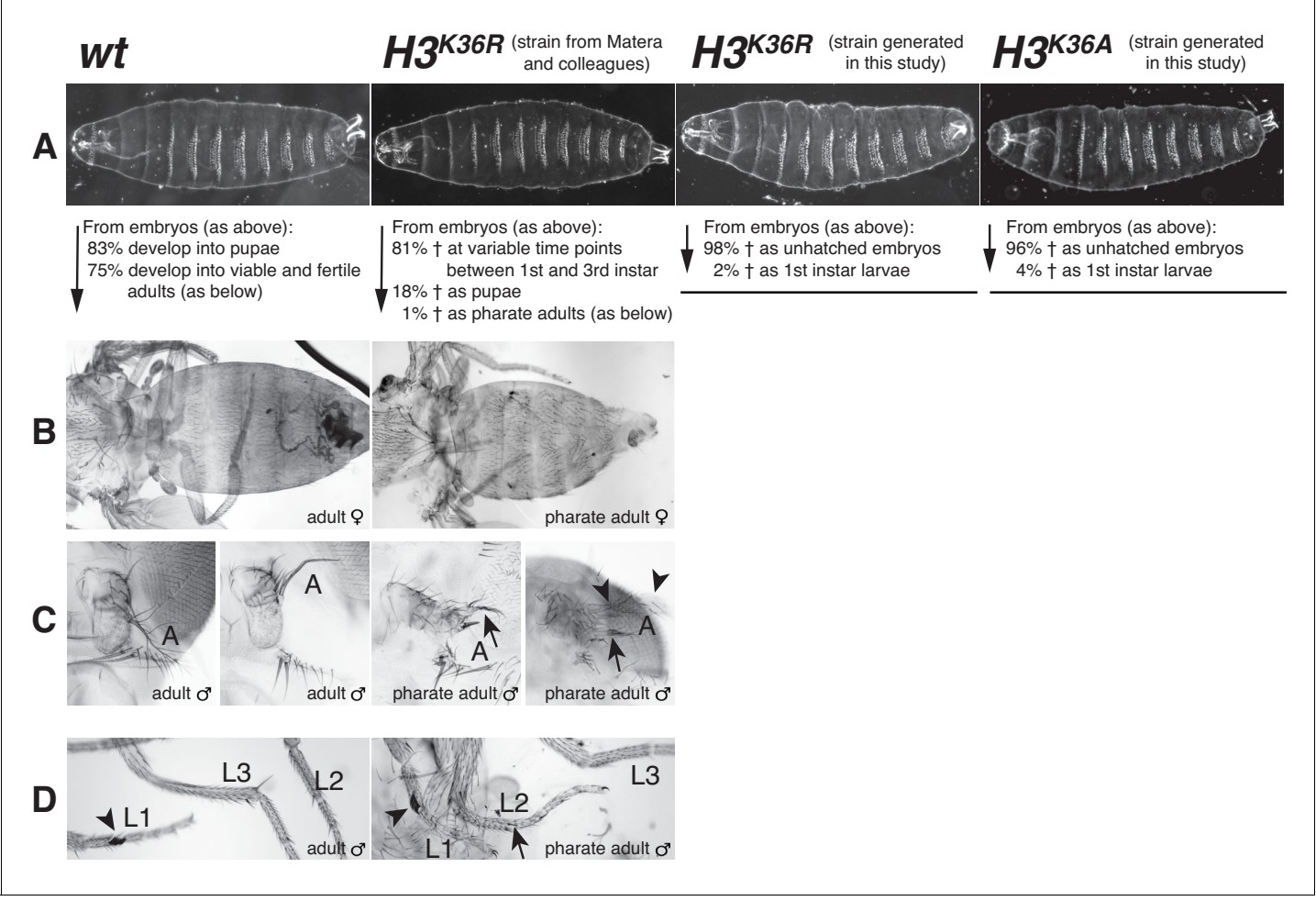

**Figure 4.** *Drosophila* with H3$^{K36R}$ or H3$^{K36A}$ mutant chromatin arrest development after completion of embryogenesis. (**A**) Ventral views of cuticles from *wildtype* (*wt*), *H3$^{K36R}$*, or *H3$^{K36A}$* mutant embryos. Note that the cuticle pattern of the mutant animals is indistinguishable from that of the *wt* embryo. Below: for each genotype, the fraction of embryos that developed into larvae, pupae, pharate adults or viable adults is listed. The fraction was determined by monitoring the development of collected hatched 1st instar larvae (*wt*: n = 300, H3$^{K36R}$ (Matera strain): n = 2000) or unhatched embryos (H3$^{K36R}$ (strain generated in this study): n = 200; H3$^{K36A}$: n = 200). The GFP marker on the Balancer chromosomes was used for identifying *H3$^{K36R}$* and *H3$^{K36A}$* mutants. See Materials and methods for further information and discussion. (**B**) Dorsal views of the posterior portion of the thorax and of the abdomen. From 2000 hatched *H3$^{K36R}$* mutant 1st instar larvae, a total of 18 pharate adults was recovered. Most *H3$^{K36R}$* mutant pharate adults showed a relatively normal overall body patterning apart from the homeotic transformations illustrated below. (**C**) Frontal view of adult heads illustrating the antenna-to-leg transformation in *H3$^{K36R}$* mutant pharate adults. The antenna-to-leg transformation in *H3$^{K36R}$* mutant animals ranged from mild (arrows) to more extensive transformations with formation of leg-like structures such as in this extreme case (arrowheads). (**D**) The sex comb in males is normally only present on the protoracic (**L1**) legs (arrowheads). Among the *H3$^{K36R}$* mutant pharate adult males recovered (n = 13), five showed one or several extra sex comb teeth (arrow) on the meso- (**L2**) or meta-thoracic (**L3**) legs. Extra sex comb teeth in adults are a hallmark phenotype of polycomb mutants.

1% developed into late pupae that complete metamorphosis but then arrested as pharate adults (*Figure 4*). Like Matera and colleagues (*McKay et al., 2015*), we have not observed any *H3$^{K36R}$* mutants that eclose from the pupal case, and both our studies therefore disagree with a report from the Schwartz lab who claimed that *H3$^{K36R}$* mutants would be able to develop into adults (*Dorafshan et al., 2019*). When we dissected the rare *H3$^{K36R}$* mutant pharate adults from their pupal cases and examined their epidermal structures, we found that they consistently showed homeotic transformations reminiscent of polycomb group (PcG) mutants. These PcG mutant phenotypes included antenna-to-leg transformations and extra sex comb teeth on meso- and meta-thoracic legs in males (*Figure 4*). The molecular analysis of these PcG phenotypes will be presented below.

The $H3^{K36R}$ mutant animals from the strain constructed in this study (i.e. containing four copies of the $3xHisGU^{H3K36R}$ miniarray) also completed embryogenesis, and their cuticle morphology was also indistinguishable from *wildtype* (*Figure 4*). However, 98% of individuals arrested development already at the end of embryogenesis and the 2% of mutant animals that hatched from the eggshell arrested development as first instar larvae (*Figure 4*). $H3^{K36A}$ mutants, containing 4 copies of a $3xHisGU^{H3K36A}$ miniarray, also completed embryogenesis and the morphology of their embryonic cuticle also appeared indistinguishable from *wildtype* (*Figure 4*). 96% of these $H3^{K36A}$ mutant animals arrested development before hatching from the eggshell and the 4% that hatched died during the first larval instar (*Figure 4*). The difference in the lethality phase of the $H3^{K36R}$ and $H3^{K36A}$ mutants generated in this study compared to $H3^{K36R}$ mutants in the strain from Matera and colleagues is possibly linked to the histone rescue transgene system used (see Materials and Methods).

## *Drosophila* with H3$^{K36R}$ or H3$^{K36A}$ mutant chromatin show diminished H3K27me3 levels at canonical PcG target genes

We next performed western blot analyses to examine H3K36me2, H3K36me3, and H3K27me3 bulk levels in $H3^{K36R}$ and $H3^{K36A}$ mutant animals. We used total nuclear extracts from late-stage $H3^{K36A}$ or $H3^{K36R}$ mutant embryos and, in the case of the $H3^{K36R}$ strain obtained from Matera and colleagues, we also used extracts from diploid imaginal disc and central nervous system (CNS) tissues dissected from surviving third instar larvae. For the interpretation of the following experiments, it is important to keep in mind that $H3^{K36R}$ and $H3^{K36A}$ zygotic mutant animals initially also contain a pool of maternally-deposited wild-type canonical H3 molecules that, together with $H3^{K36R}$ and $H3^{K36A}$, become incorporated into chromatin during the pre-blastoderm cleavage cycles, up to and including the S-phase of cell cycle 14. It is only from the S-phase of cell cycle 15 onwards when only transgene-encoded histones then become incorporated into chromatin (*Günesdogan et al., 2010*). Although the wild-type H3 molecules in chromatin become diluted during every cell cycle and are eventually fully replaced by mutant H3, they are probably still present in the chromatin of late-stage embryos. The effective replacement of persisting wild-type H3 molecules by mutant H3 greatly varies between embryonic tissues because of the different numbers of cell divisions that take place in the different tissues prior to the end of embryogenesis. For example, whereas epidermal cells undergo only two more divisions after S-phase 14, certain cells in the CNS undergo as many as 12 divisions (*Bossing et al., 1996*). In diploid tissues from $H3^{K36R}$ mutant larvae, replacement of wild-type H3 by H3$^{K36R}$ can be expected to be much more complete because of the extensive cell proliferation that occurs in these tissues that, after metamorphosis, will give rise to the structures of the adult body.

We first performed western blot analyses on $H3^{K36R}$ mutant larvae and found that H3K36me2 and H3K36me3 bulk levels were reduced more than 4-fold compared to *wildtype* (*Figure 5A*). The residual H3K36me2 and H3K36me3 signals (*Figure 5A*, lane 4) probably represent the methylated versions of the histone variant H3.3 that are encoded by the genes *H3.3A* and *H3.3B* that are not located in the *HisC* locus and had not been mutated in these animals. Intriguingly, $H3^{K36R}$ mutant animals also showed an about two-fold reduction in H3K27me3 bulk levels compared to *wildtype* (*Figure 5A*, compare lanes 4–6 with 1–3). The reduction of not only H3K36me2 and H3K36me3 but also of H3K27me3 bulk levels in $H3^{K36R}$ mutant larvae had previously also been noted by Matera and colleagues (*Meers et al., 2017*). H3K27 tri-methylation by PRC2 therefore appears to be compromised in *Drosophila* chromatin consisting of H3$^{K36R}$ nucleosomes.

We then did western blot analyses on extracts from $H3^{K36A}$ and $H3^{K36R}$ mutant embryos obtained from the strains generated in this study. In $H3^{K36A}$ mutants, H3K36me2 and H3K36me3 bulk levels were reduced about 3- to 4-fold compared to *wildtype* (*Figure 5B*, compare lanes 5–8 with 1–4). In $H3^{K36R}$ mutants, H3K36me2 and H3K36me3 bulk levels unexpectedly appeared much less severely reduced (*Figure 5B*, compare lanes 9–12 with 1–4 and 5–8). As discussed above, the residual H3K36me2 and H3K36me3 signal in $H3^{K36A}$ and $H3^{K36R}$ mutant embryos might in part represent modified maternally-deposited canonical wild-type H3, and in part the modified H3.3 variants. However, the reason for the differential reduction of H3K36me2 and -me3 levels in $H3^{K36A}$ and $H3^{K36R}$ mutant embryos remains unclear. In both genotypes, H3K27me3 bulk levels appeared largely unchanged compared to *wildtype* (*Figure 5B*, compare lanes 5–8 and 9–12 with 1–4).

We next performed ChIP-seq experiments to examine how the genome-wide profiles of H3K36me2 and H3K27me3 are changed in $H3^{K36R}$ and $H3^{K36A}$ mutants. In the case of $H3^{K36R}$

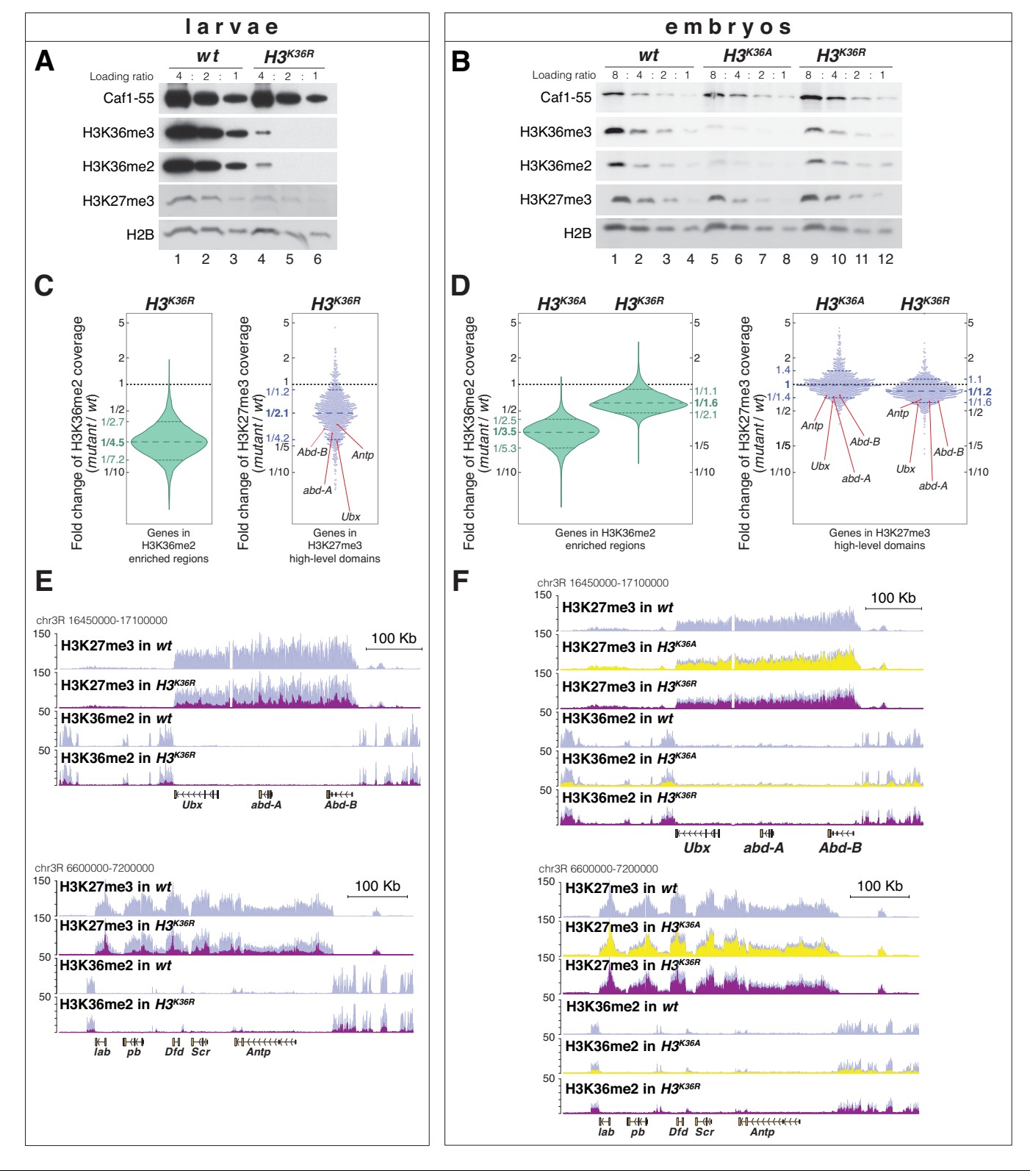

**Figure 5.** $H3^{K36A}$ and $H3^{K36R}$ mutants show reduced levels of H3K27me3. (**A**) Western blot analysis on serial dilutions (4:2:1) of total cell extracts from wing, haltere and 3rd leg imaginal disc tissues dissected from *wildtype* (*wt*, lanes 1–3) and $H3^{K36R}$ mutant (lanes 4–6) third instar larvae. Blots were probed with antibodies against H3K36me3, H3K36me2, or H3K27me3; in each case, probing of the same membranes with antibodies against Caf1-55 and H2B served as controls for loading and western blot processing. Note the reduced levels of H3K36me3 and H3K36me2 but also of H3K27me3 in

*Figure 5 continued on next page*

*Figure 5 continued*

$H3^{K36R}$ mutants compared to *wildtype* (*wt*) (see text). See Materials and Methods for details of all genotypes. (B) Western blot analysis on serial dilutions (8:4:2:1) of total nuclear extracts from 21 to 24 hr old *wt* (lanes 1–4), $H3^{K36A}$ mutant (lanes 5–8) and $H3^{K36R}$ mutant (lanes 9–12) embryos, probed with antibodies against H3K36me3, H3K36me2 or H3K27me3; and with antibodies against Caf1-55 and H2B as controls. Note that H3K36me3 and H3K36me2 levels are reduced in $H3^{K36A}$ mutants but not in $H3^{K36R}$ mutants where they are comparable to *wt*. Also note that H3K27me3 levels appear undiminished in either mutant (see text). (C) Left, violin plot showing the fold-change of H3K36me2 coverage in $H3^{K36R}$ mutant larvae relative to *wt* at genes that in *wildtype* larval CNS and imaginal disc tissues are decorated with H3K36me2 (see Materials and Methods). The dashed line marks the median reduction (4.5-fold), the dotted lines indicate the interval comprising 80% of regions. Right, Bee plot showing the fold-change of H3K27me3 coverage in $H3^{K36R}$ mutant larvae relative to *wt* at genes that in *wildtype* larval CNS and imaginal disc tissues are associated with high-level H3K27me3 regions (see Materials and Methods). The dashed line marks the median reduction (2.1-fold), the dotted lines indicate the interval comprising 80% of regions. Note that H3K27me3 coverage at the HOX genes *abd-A*, *Abd-B*, *Ubx* and *Antp* is between 3- and 4-fold reduced. (D) Analysis and representation as in (C) but showing fold-changes in H3K36me2 and H3K27me3 coverage in $H3^{K36A}$ and $H3^{K36R}$ mutant late-stage embryos relative to *wt* at genes that in *wildtype* embryos are decorated with H3K36me2 and H3K27me3, respectively. Note that H3K27me3 coverage at the HOX genes *abd-A*, *Abd-B*, *Ubx* and *Antp* is about 1.5-fold reduced. See also *Figure 5—figure supplement 1*. (E) H3K27me3 and H3K36me2 ChIP-seq profiles in larval CNS and imaginal disc tissues from *wt* (blue) and $H3^{K36R}$ mutant (purple) third instar larvae; in the tracks showing the profiles in the $H3^{K36R}$ mutant, the *wt* profile is superimposed as reference (see *Supplementary file 2* and Materials and Methods for information about normalization). Top: genomic interval containing the *Bithorax-Complex* harboring the HOX genes *Ubx*, *abd-A* and *Abd-B*; bottom: genomic interval containing the *Antennapedia-Complex* with the HOX genes *lab*, *pb*, *Dfd*, *Scr*, and *Antp*. Note the 3- to 4-fold reduction of H3K27me3 levels across the *Bithorax* and *Antennapedia* loci in $H3^{K36R}$ mutants. Also note that for every HOX gene, the analyzed tissues (CNS, thoracic imaginal discs and eye-antenna discs) represent a mixed population of cells with a fraction of cells in which the gene is inactive, decorated with H3K27me3 and repressed by PcG and fraction of cells in which the gene is transcriptionally active and carrying the H3K36me2 modification. (F) H3K27me3 and H3K36me2 ChIP-seq profiles at the Bithorax and Antennapedia loci as in (E) but from *wt* (blue), $H3^{K36A}$ mutant (yellow) and $H3^{K36R}$ mutant (purple) late-stage embryos with the *wt* profile superimposed in the tracks showing the profiles in the $H3^{K36A}$ and $H3^{K36R}$ mutants. H3K27me3 levels across the Bithorax and Antennapedia loci in $H3^{K36A}$ and $H3^{K36R}$ mutants are only about 1.5-fold reduced compared to *wt*.

The online version of this article includes the following figure supplement(s) for figure 5:

**Figure supplement 1.** $H3^{K36R}$ and $H3^{K36A}$ mutants show altered H3K36me2 and H3K27me3 profiles (related to *Figure 5*).

mutants, we compared these profiles in cells from imaginal disc and CNS tissues dissected from late-stage third instar $H3^{K36R}$ and *wildtype* larvae. In parallel, we also compared the profiles in late-stage $H3^{K36A}$, $H3^{K36R}$ and *wildtype* embryos, for both mutants using the strains generated in this study. As expected from the western blot analyses (*Figure 5A*), H3K36me2 levels across the genome were strongly diminished in $H3^{K36R}$ mutant larvae (*Figure 5C*, *Figure 5—figure supplement 1*, *Supplementary file 2*). The genome-wide profile of H3K36me2 was also substantially reduced in $H3^{K36A}$ mutant embryos but, as expected from the western blot analyses, the profile was less severely reduced in $H3^{K36R}$ mutant embryos (*Figure 5D*, *Figure 5—figure supplement 1*, *Supplementary file 2*).

The H3K27me3 genomic profile confirmed that the levels of this modification were considerably reduced in the chromatin of late-stage $H3^{K36R}$ mutant larvae (*Figure 5C*). While the average reduction was only about two-fold (*Figure 5C*), H3K27me3 levels were particularly strongly diminished at canonical PRC2 target genes such as the HOX genes that in *wildtype* animals are decorated with high-levels of H3K27me3 (*Figure 5C,E*, *Figure 5—figure supplement 1*, *Supplementary file 2*). Specifically, at the HOX genes *Ultrabithorax* (*Ubx*), *abdominal-A* (*abd-A*), *Abdominal-B* (*Abd-B*) or *Antennapedia* (*Antp*), H3K27me3 levels in $H3^{K36R}$ mutants were between three and fourfold lower than in *wildtype* (*Figure 5C,E*).

As expected from the western blot analyses (*Figure 5B*), $H3^{K36A}$ or $H3^{K36R}$ mutant embryos showed no general reduction in their genome-wide H3K27me3 profiles (*Figure 5D*). However, in both mutants, H3K27me3 levels were about 1.5-fold reduced across the HOX gene loci (*Figure 5D, F*). In *Drosophila* with $H3^{K36R}$ or $H3^{K36A}$ chromatin, PRC2 therefore appears to be unable to generate high levels of H3K27me3 at Polycomb target genes.

## Polycomb repression of HOX genes is impaired in *Drosophila* with $H3^{K36R}$ or $H3^{K36A}$ mutant chromatin

The PcG-like phenotypes in the rare $H3^{K36R}$ mutant animals that survive into pharate adults and the reduction of H3K27me3 levels in HOX gene chromatin in these mutants prompted us to analyze whether and how expression of these genes is altered in $H3^{K36R}$ and $H3^{K36A}$ mutants. In a first set of experiments, we analyzed HOX gene expression in embryos. Both mutants showed stochastic

misexpression of *Abd-B* in single cells or pairs of cells in the CNS of late-stage embryos (*Figure 6A*). *Abd-B* misexpression in $H3^{K36R}$ and $H3^{K36A}$ mutant embryos was however clearly less widespread than in $H3^{K27R}$ mutant embryos or in embryos lacking the PRC2 subunit Esc that are shown for comparison (*Figure 6A*). Moreover, we were unable to detect misexpression of *Antp* or *Ubx* in $H3^{K36R}$ or $H3^{K36A}$ mutant embryos.

We next analyzed HOX gene expression in imaginal discs and CNS tissues from third instar $H3^{K36R}$ mutant larvae. In the CNS of every single mutant individual, *Ubx* was widely misexpressed in many single cells in an apparently stochastic pattern (*Figure 6B*). 50% of the $H3^{K36R}$ mutant larvae also showed stochastic misexpression of *Ubx* in individual cells in the wing blade primordium of the wing imaginal disc (*Figure 6C*), the area of this disc where Ubx is most readily de-repressed if PcG function is perturbed (*Beuchle et al., 2001*). *Ubx* misexpression in $H3^{K36R}$ mutant wing discs was less widespread than in clones of $H3^{K27R}$ mutant cells that were induced in $H3^{K27R}$ heterozygotes and are shown for comparison (*Figure 6C*). Finally, we found that 100% of the $H3^{K36R}$ mutant larvae showed misexpression of *Antp* in the antenna primordium of the eye-anntennal disc (*Figure 6D*). We also observed this misexpression in clones of $H3^{K36A}$ homozygous cells that we had induced in $H3^{K36A}$ heterozygous animals (*Figure 6D*) and in $H3^{K27R}$ mutant clones that were induced as control (*Figure 6D*). Analogous to *Ubx* in the wing disc, *Antp* is misexpressed in the antenna primordium, the region of the eye-antenna imaginal discs where Antp is most susceptible to becoming misexpressed if PcG function is compromised. In this context, it should also be emphasized that the reduction of H3K27me3 signal in tissues from third instar $H3^{K36R}$ mutant larvae is quite uniform across the tissues, as illustrated in *Figure 6—figure supplement 1*. It should also be emphasized that transcriptome analyses on whole $H3^{K36R}$ mutant third instar larvae found no extensive global deregulation of gene transcription (*Meers et al., 2017*). The most straightforward explanation for the stochastic misexpression of multiple HOX genes in animals with chromatin consisting of $H3^{K36R}$ or $H3^{K36A}$ nucleosomes therefore is that it is caused by defective Polycomb repression as a result of the reduced H3K27me3 levels in HOX gene chromatin.

## Discussion

Understanding how PRC2 binds chromatin and how it is regulated is essential for understanding how the complex marks genes for Polycomb repression to maintain cell fate decisions. The work in this study leads to the following main conclusions. First, the structure of nucleosome-bound PHF1-PRC2 allowed us to visualize how interaction of the catalytic lobe of the complex with the substrate nucleosome threads the histone H3 N-terminus into the active site of EZH2 through a relay of contacts. Second, structure-guided mutational analyses showed that DNA-binding by the $EZH2_{CXC}$ domain is critical for productive PRC2-nucleosome interactions. Third, unmodified H3K36 is accommodated in a key position in the $EZH2_{CXC}$-DNA interface and while H3K36 provides the correct fit, the methylated forms H3K36me2/3, or mutated H3K36R or H3K36A do not seem to fit because they strongly diminish H3K27 methylation. Fourth, H3K36 is also critical for normal H3K27 methylation *in vivo* because *Drosophila* with $H3^{K36R}$ or $H3^{K36A}$ mutant chromatin show reduced levels of H3K27me3 and fail to fully maintain Polycomb repression at HOX target genes. In the following, we shall discuss key aspects of these new findings in the context of our previous knowledge of PRC2 regulation and function.

### Different forms of PRC2 use the same molecular interactions for binding the H3 N-terminus on substrate nucleosomes

Unlike many other histone-modifying enzymes (e.g. *McGinty et al., 2014*; *Worden et al., 2019*), PRC2 does not recognize the nucleosome by docking on its acidic patch (*Luger et al., 1997*) to engage with the histone substrate. Instead, the complex interacts with chromatin by binding to the DNA gyres on the nucleosome (*Poepsel et al., 2018*, this study). Prevous studies that had measured the binding affinity and residence time of PRC2 on nucleosomes and free DNA had found that DNA-binding makes the largest contribution to the chromatin-binding affinity of PRC2 (*Choi et al., 2017*; *Wang et al., 2017*). The mutational analyses here establish that interaction of highly conserved residues in the $EZH2_{CXC}$ domain with the DNA on the substrate nucleosome is critical for H3K27 methylation (*Figure 2C*). Moreover, this interaction sets the register for a network of interactions of the H3 N-terminus with the EZH2 surface that permits H3K27 to reach into the active site (*Figure 1D,F*).

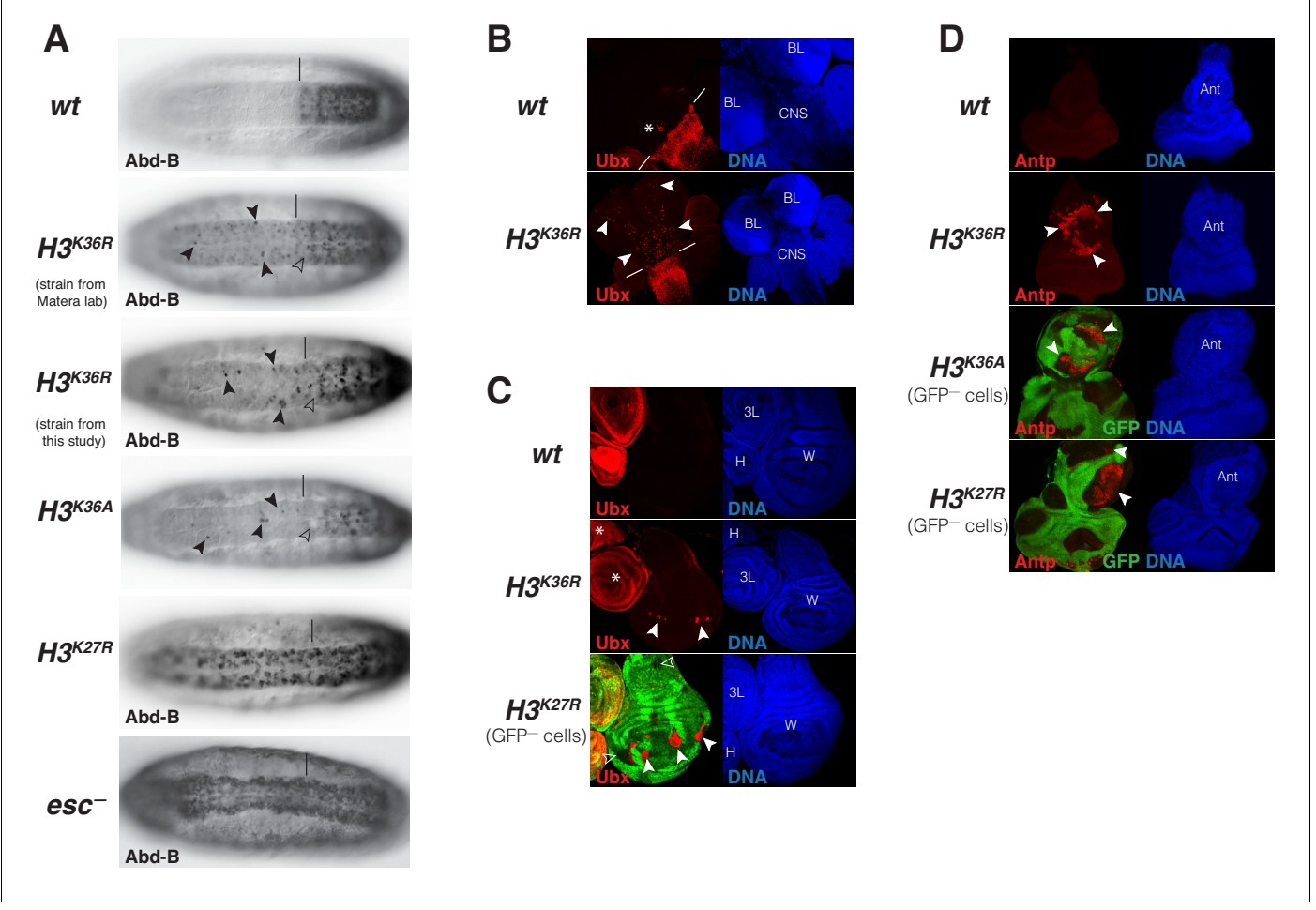

**Figure 6.** *Drosophila* with H3[K36R] or H3[K36A] chromatin show defective Polycomb repression at HOX genes. (**A**) Ventral views of stage 16 *wildtype* (*wt*), *H3[K36A]*, *H3[K36R]*, *H3[K27R]*, or *esc* (*esc⁻*) mutant embryos, stained with antibody against Abd-B protein; the *esc* mutant embryo lacked both maternal and zygotic expression of *esc* (see Materials and Methods for details of all genotypes). The vertical bar marks the anterior boundary of *Abd-B* expression in parasegment (ps) 10 in *wt* embryos. Note the stochastic misexpression of Abd-B protein in single cells or pairs of cells anterior to ps10 in *H3[K36R]* and *H3[K36A]* mutant embryos (arrowheads). *H3[K27R]* and *esc* mutant embryos show widespread misexpression of Abd-B protein in the head-to-tail pattern characteristic of PcG mutants. For reasons that are not well understood, *H3[K36A]* and *H3[K36R]* mutants also show partial loss of Abd-B expression in cells in ps10 (empty arrowheads). (**B**) Larval CNS and brain lobe tissues from *wildtype* (*wt*) or *H3[K36R]* mutant third instar larvae, stained with antibody against Ubx protein (red) and Hoechst (DNA) to label all nuclei; location of CNS and brain lobes (BL) are indicated in the right panel. The white bars mark the anterior boundary of *Ubx* expression in ps5 in *wt* embryos, the asterisk marks the Ubx-expressing cells in the central midline of ps4 that are part of the wild-type Ubx pattern. Note the stochastic misexpression of Ubx protein in many single cells anterior to ps5 in the CNS and in the brain lobes (arrowheads). (**C**) Imaginal wing (**W**), haltere (**H**) and 3rd leg (3L) discs from *wildtype* (*wt*) or *H3[K36R]* mutant third instar larvae and, as reference, discs from a larvae with clones of *H3[K27R]* mutant cells that are marked by the absence of GFP. In all cases, discs were stained with antibody against Ubx protein (red) and Hoechst (DNA) to label all nuclei. In *wt* animals, Ubx is expressed in the haltere and 3rd leg disc but not in the wing disc where it is repressed by the PcG machinery. Note that in *H3[K36R]* mutants, Ubx is misexpressed in small clusters of cells in the wing blade primordium of the wing disc (arrowheads) but remains repressed in the rest of the wing disc. Such misexpression was detected in 50% of wing discs (*n* = 28). As reference, a wing discs with *H3[K27R]* mutant clones is shown, where all cells in the clones in the wing blade primordium (arrowheads) show misexpression of Ubx whereas cells in the notum and hinge primordium show no misexpression (empty arrowheads) (cf. *Pengelly et al., 2013*). Also note that in *H3[K36R]* mutants (*n* > 30 mutant animals analyzed), Ubx expression in haltere and leg discs appears unperturbed (asterisks). (**D**) Eye-antennal imaginal discs from *wildtype* (*wt*) or *H3[K36R]* mutant larvae and below discs from larvae with clones of *H3[K36A]* or *H3[K27R]* mutant cells that are marked by the absence of GFP. All animals were stained with antibody against Antp protein (red) and Hoechst (DNA) to label all nuclei. Antp is not expressed in the eye-antennal disc of *wt* animals. Note that in *H3[K36R]* mutant discs, Antp is misexpressed in large clusters of cells (arrowheads) in the antenna primordium (Ant). Note that Antp is also misexpressed in *H3[K36A]* or *H3[K27R]* mutant cell clones in the antenna primordium (arrowheads) and that in these cases misexpression also only occurs in a subset of the mutant cells and not in all clones.

The online version of this article includes the following figure supplement(s) for figure 6:

**Figure supplement 1.** H3K27me3 levels are uniformly reduced in imaginal disc tissues from *H3[K36R]* mutant larvae (related to *Figure 6*).

Consistent with our findings here, an independent recent study of a cryo-EM structure of PRC2 with co-factors JARID2 and AEBP2 bound to a mononucleosome with monoubiquitylated H2A (*Kasinath et al., 2020*) identified very similar interactions of EZH2 with the nucleosomal DNA and the H3 N-terminus. Different forms of PRC2 that contain different accessory proteins and dock in different ways on chromatin therefore contact the substrate H3 N-terminus in the nucleosome through similar interactions.

## The position of H3K36 in the EZH2$_{CXC}$-nucleosome interface enables allosteric regulation by H3K36 methylation

Important novel insight from our structure came from the observation that unmodified H3K36 is located in a critical position in the EZH2$_{CXC}$-DNA interface. Unmodified H3K36 has the right fit for interaction of the H3 N-terminus with the EZH2 surface and placement of H3K27 in the active site. The inhibition of H3K27 mono-, di- and tri-methylation on nucleosomes carrying H3K36me2 or -me3 (*Schmitges et al., 2011*; *Yuan et al., 2011*) or on H3$^{K36R}$ or H3$^{K36A}$ nucleosomes (*Figure 3C*) suggests that even though these side chains could theoretically be accommodated in the EZH2$_{CXC}$-DNA interface, these alterations of the side chain of residue 36 in H3 chain must somehow impair productive interaction of H3K27 with the catalytic center of EZH2. On isolated H3 N-terminal peptides, H3K36me3 did not inhibit the formation of H3K27me1 (*Figure 3D*), consistent with earlier findings that on peptide substrates H3K36me3 only has a minor effect on the k$_{cat}$ of H3K27 methylation (*Schmitges et al., 2011*; *Jani et al., 2019*). Also, H3K36me3 does not diminish the affinity of PRC2 for binding to mononucleosomes (*Figure 3A,B*) and does not reduce the residence time of PRC2 on nucleosome arrays (*Guidotti et al., 2019*). Taken together, a possible scenario would therefore be that within the time frame of the PRC2-nucleosome binding and reaction cycle, docking of the H3K36 side chain in the EZH2$_{CXC}$-DNA interface is critical for rapid alignment of the H3 N-terminus on the EZH2 surface into a catalytically competent state. According to this view, H3K36me2/3 does not locally disrupt nucleosome binding but allosterically inhibits H3K27 from interacting with the EZH2 active site.

## H3K27 methylation and polycomb repression are defective in *Drosophila* with H3$^{K36R}$ or H3$^{K36A}$ chromatin

The finding that PRC2 is inhibited on H3$^{K36R}$ and H3$^{K36A}$ nucleosomes *in vitro* had prompted us to use a genetic histone replacement strategy in *Drosophila* (*Günesdogan et al., 2010*; *McKay et al., 2015*) to assess PRC2 inhibition on H3$^{K36R}$ or H3$^{K36A}$ chromatin *in vivo*. Previous studies had found that *Drosophila* H3$^{K36R}$ mutants are able to develop into the pupal stages and, consistent with this late developmental arrest, whole third instar larvae were found to show only relatively minor changes in their transcriptome compared to *wildtype* animals (*McKay et al., 2015*; *Meers et al., 2017*). Of relevance here, these transcriptome analyses did not reveal any gross deregulation of HOX or PcG genes (*Meers et al., 2017*). Here, we found that a few rare *H3$^{K36R}$* mutant animals even survive into pharate adults and that these show remarkably little morphological defects apart from homeotic transformations characteristic of Polycomb mutants (*Figure 4*). We show that these phenotypes are caused by misexpression of multiple HOX genes (*Figure 6*). HOX gene misexpression in *H3$^{K36R}$* or *H3$^{K36A}$* mutants is stochastic and not as widespread as in strong PcG mutants but it occurs in cells and tissues where HOX genes also first become misexpressed if PcG function is removed. We found that HOX misexpression is directly linked to reduced levels of H3K27me3 at these genes (*Figure 5*). A simple straightforward explanation for these phenotypes in *H3$^{K36R}$* or *H3$^{K36A}$* mutant animals is that PRC2 is unable to effectively deposit high levels of H3K27me3 on the H3$^{K36R}$ or H3$^{K36A}$ nucleosomes, respectively, in their chromatin. Accordingly, H3K27me3 levels at HOX genes are below the threshold needed to stringently maintain Polycomb repression and consequently, HOX genes become stochastically misexpressed in a fraction of cells. Finally, we note that in *H3$^{K36R}$* mutant larvae, the experimental setting where we have been able to generate the most complete replacement of H3 by H3$^{K36R}$, H3K27me3 levels at HOX genes were only about 3- to 4-fold reduced compared to *wildtype* (*Figure 5C*). However, as shown in *Figure 3C*, on nucleosomes *in vitro*, H3K36me3 inhibited PRC2 more effectively than H3$^{K36R}$ or H3$^{K36A}$. It therefore seems likely that in contrast to the *H3$^{K36R}$* and *H3$^{K36A}$* mutants that we have used as proxy, H3K36me2 and H3K36me3 *in vivo* also

inhibit PRC2 more effectively from depositing H3K27me3 on H3K36me2- or H3K36me3-modified nucleosomes in transcriptionally active chromatin.

## Concluding remark

The structural, biochemical and genetic work reported in this study shows that it is the exquisite geometry formed by a relay of interactions between the PRC2 enzyme, nucleosomal DNA and the H3 N-terminus that enable the histone methylation marks H3K36me2 and H3K36me3 in transcriptionally active chromatin to allosterically prevent PRC2 from depositing the repressive histone methylation mark H3K27me3 at transcribed genes.

# Materials and methods

## Key resources table

| Reagent type (species) or resource | Designation | Source or reference | Identifiers | Additional information |
|---|---|---|---|---|
| Strain, strain background *D. melanogaster* | *Oregon-R* | Flybase | | |
| Strain, strain background *D. melanogaster* | *w; Df(2L)His$^C$ FRT40A/Df(2L)His$^C$ FRT40A; 12xHisGU$^{wt}$/12 xHisGU$^{wt}$* | **McKay et al., 2015** | | |
| Strain, strain background *D. melanogaster* | *w; Df(2L)His$^C$ FRT40A/CyO ubi-GFP; 12xHisGU$^{H3K36R}$/TM6B* | **McKay et al., 2015** | | |
| Strain, strain background *D. melanogaster* | *w; Df(2L)His$^C$ FRT40A/CyO twi:Gal4 UAS:GFP; 3xHisGU$^{H3K36A}$(VK33) 3xHisGU$^{H3K36A}$(86Fb) /3xHisGU$^{H3K36A}$(VK33) 3xHisGU$^{H3K36A}$(86Fb)* | This study | | Available on request |
| Strain, strain background *D. melanogaster* | *w; Df(2L)His$^C$ FRT40A/CyO twi:Gal4 UAS:GFP; 3xHisGU$^{H3K36R}$(VK33) 3xHisGU$^{H3K36R}$(86Fb) /3xHisGU$^{H3K36R}$(VK33) 3xHisGU$^{H3K36R}$(86Fb)* | This study | | Available on request |
| Strain, strain background *D. melanogaster* | *w; Df(2L)His$^C$ FRT40A/CyO ubi:GFP; 3xHisGU$^{H3K27R}$(68E) 3xHisGU$^{H3K27R}$ (86Fb)/3xHis GU$^{H3K27R}$ (68E) 3xHisGU$^{H3K27R}$ (86Fb)* | **Pengelly et al., 2013** | | |
| Strain, strain background *D. melanogaster* | *w hs-flp; w; hs-nGFP FRT40A/hs nGFP FRT40; 3xHisGU$^{H3K27R}$(68E) 3x HisGU$^{H3K27R}$(86Fb) /3xHisGU$^{H3K27R}$(68E) 3xHisGU$^{H3K27R}$ (86Fb)* | **Pengelly et al., 2013** | | |
| Strain, strain background *D. melanogaster* | *w hs-flp; M(2)25A ubi-GFP FRT40A/CyO* | Müller lab stocks | | Available on request |
| Strain, strain background *D. melanogaster* | *yw; esc$^6$ b pr/CyO, P[esc$^+$]* | Struhl laboratory **Struhl, 1981** | | |
| Strain, strain background *D. melanogaster* | *In(2LR) Gla/CyO, esc$^2$* | Struhl laboratory **Struhl, 1981** | | |
| Strain, strain background *D. melanogaster* | *w hs-flp; hs-nGFP FRT2A/hs nGFP FRT2A* | **Beuchle et al., 2001** | | |
| Strain, strain background (*Escherichia coli*) | BL21(DE3) | Sigma-Aldrich | CMC0016 | Electrocompetent cells |

*Continued on next page*

*Continued*

| Reagent type (species) or resource | Designation | Source or reference | Identifiers | Additional information |
|---|---|---|---|---|
| Cell line (*Trichoplusia ni*) | HighFive cell line for expression | Invitrogen | Product nr.: B85502 BTI-Tn-5B1-4 (RRID:CVCL_C190) | Protein expression |
| Cell line (*Spodoptera frugiperda*) | Sf21 cell line for Baculovirus production | Invitrogen | Product nr.: 1149701 (RRID:CVCL_0518) | Baculovirus production for protein expression |
| Antibody | H3K27me3 Rabbit monoclonal antibody | Cell Signaling Technology | Cell Signaling Technology #9733 | IF (1:50) WB (1:2000) ChIP (1:500) |
| Antibody | H3K27me3 Rabbit polyclonal antibody | Millipore | Millipore #07–449 | WB (1:1000) |
| Antibody | H3K27me1 Rabbit polyclonal antibody | Millipore | Millipore #07–448 | WB (1:6000) |
| Antibody | H3K36me3 Rabbit monoclonal antibody | Cell Signaling Technology | Cell Signaling Technology #4909 | WB (1:750) |
| Antibody | H3K36me2 Rabbit monoclonal antibody | Cell Signaling Technology | Cell Signaling Technology #2901 | WB (1:250) |
| Antibody | H3K36me2 Rabbit monoclonal antibody | Abcam | #9049 | ChIP (1:300) |
| Antibody | H2B Rabbit polyclonal antibody | This study | Raised against full-length recombinant *Drosophila* H2B | WB (1:10000) Available on request |
| Antibody | H4 Rabbit polyclonal antibody | Abcam | Abcam #10158 | WB (1:200000) |
| Antibody | Caf1 Rabbit polyclonal antibody | *Gambetta et al., 2009* | Müller lab | WB (1:10000) |
| Antibody | Abd-B Mouse monoclonal antibody | DHSB | DSHB (1A2E9) | IF (1:300) |
| Antibody | Antp Mouse monoclonal antibody | DHSB | DSHB (8C11) | IF (1: 100) |
| Recombinant DNA reagent | *pfC31-attB-3xHisGU.H3K36A* | This study | | See Materials and Methods Available on request |
| Recombinant DNA reagent | *pfC31-attB-3xHisGU.H3K36R* | This study | | See Materials and Methods Available on request |
| Recombinant DNA reagent | nucleosome-positioning sequence 601 (147 bp + linker version) | *Lowary and Widom, 1998*, *Nekrasov et al., 2005* | | |
| Recombinant DNA reagent | pFB-EZH2 | *Choi et al., 2017* | | N-terminal 6xHis-tag |
| Recombinant DNA reagent | pFB-EZH2$^{CXC>A}$ | This study | | N-terminal 6xHis-tag See Materials and Methods Available on request |
| Recombinant DNA reagent | pFB-EED | *Choi et al., 2017* | | N-terminal 6xHis-tag |

*Continued on next page*

*Continued*

| Reagent type (species) or resource | Designation | Source or reference | Identifiers | Additional information |
|---|---|---|---|---|
| Recombinant DNA reagent | pFB-EED$^{EED>A}$ | This study | | N-terminal 6xHis-tag See Materials and Methods Available on request |
| Recombinant DNA reagent | pFB-SUZ12 | *Choi et al., 2017* | | N-terminal 6xHis-tag |
| Recombinant DNA reagent | pFB-RBBP4 | *Choi et al., 2017* | | N-terminal 6xHis-tag |
| Recombinant DNA reagent | pFB-PHF1 | *Choi et al., 2017* | | N-terminal twin-strep and 6xHis-tag (SHT) |
| Recombinant DNA reagent | pFB-PHF1$_C$ (PHF1$_{515-567}$) | *Choi et al., 2017* | | N-terminal twin-strep and 6xHis-tag (SHT) |
| Peptide, recombinant protein | H3$_{18-42}$ peptide | MPIB core facilty | | |
| Peptide, recombinant protein | H3$_{18-42}$K36me3 peptide | MPIB core facilty | | |

## Protein expression and purification

Human PHF1-PRC2 wild-type (wt) complex was expressed and purified as previously described (*Choi et al., 2017*). In brief, an optimized ratio of the baculoviruses (produced in Sf21 cells, (Invitrogen 1149701)) for the different PHF1-PRC2 subunits was used to infect *Trichoplusia ni* High Five insect cells (Invitrogen B85502). The Sf21 and High Five cells were authenticated by genotyping (Eurofins) and tested negative for mycoplasma contamination (LookOut Mycoplasma PCR Detection Kit, Sigma-Aldrich). Cells were lysed using a glass Dounce homogenizer and the complex was purified using affinity chromatography (Ni-NTA and Strep-tag), followed by simultaneous TEV mediated protease tag cleavage and Lambda Phosphatase treatment (obtained from the MPI of Biochemistry Protein Core facility) and a final size-exclusion chromatography (SEC) step in a buffer containing 25 mM Hepes, pH 7.8, 150 mM NaCl, 10% glycerol, 2 mM DTT.

PRC2$^{CXC>A}$, PRC2$^{EED>A}$ and PRC2$^{CXC>A/EED>A}$ mutants were generated by PCR with primers containing the desired mutations, subsequent ligation and transformation. Expression and purification were performed as above.

*Xenopus laevis* (*X.l.*) and *D. melanogaster* (*D.m.*) histones were expressed in *E.coli* strains BL21 and purified from inclusion bodies as described in *Luger et al., 1999*. To mimic the inhibitory mark H3K36me3 or the allosteric activating mark H3K27me3, the cysteine side chain of a mutated *D.m.* histone H3$^{C110A\ K36C}$ or *X.l.* histone H3$^{C110A\ K27C}$ was alkylated with (2-bromoethyl) trimethylammonium bromide (Sigma-Aldrich) as described previously (*Simon et al., 2007*). Nucleosomes containing these modifications are abbreviated with e.g. H3Kc36me3.

For histone octamers, equimolar amounts of histones H2A, H2B, H4 and H3 (wt, H3$^{K36A}$, H3$^{K36R}$, H3Kc27me3 or H3Kc36me3) were mixed and assembled into octamers in high salt buffer containing 10 mM Tris-HCL pH 7.5, 2 M NaCl, 1 mM EDTA, 5 mM β-mercaptoethanol. Subsequent SEC was performed to separate octamers from H3/H4 tetramers or H2A/H2B dimers (*Luger et al., 1999*).

## Reconstitution of nucleosomes

For *X.l.* and *D.m* mononucleosomes used in biochemical assays, 6-carboxyfluorescein (6-FAM)-labeled 215 bp 601 DNA (*Lowary and Widom, 1998*) was PCR amplified from the p601 plasmid, purified on a MonoQ column (GE Healthcare), precipitated with ethanol and dissolved in the same high salt buffer used for octamers. Optimized ratios of octamer to DNA (usually ranging between 0.8–1.3: 1) were mixed and nucleosomes were reconstituted by gradient and stepwise dialysis against low salt buffers to a final buffer containing 25 mM Hepes, pH 7.8, 60 mM NaCl, 2 mM DTT.

*X.l.* asymmetrical dinucleosomes for cryo-EM studies containing one unmodified substrate nucleosome and one H3Kc27me3-modified (allosteric) nucleosome connected with a 35 bp linker DNA were reconstituted using the protocol described in *Poepsel et al., 2018*. In brief, substrate nucleosomes and allosteric nucleosomes were separately assembled on the respective *DraIII* digested nucleosomal DNA. The latter was generated by PCR with primers introducing the desired linker and *DraIII* recognition sites and purified as described above. The assembled nucleosomes were purified

on a preparative native gel system (Biorad 491 prep cell). After ligation using T4 ligase (Thermo Fisher Scientific) the resulting dinucleosomes were purified from aberrant or non-ligated mononucleosomes by a second preparative native gel system (Biorad 491 prep cell). In contrast to *Poepsel et al., 2018*, the dinucleosome DNA used in this study contained an additional 30 bp overhang on the substrate nucleosome, thus resulting in the following DNA sequence:

5'–601 binding (allosteric nucleosome) – agcgatctCACCCCGTGatgctcgatactgtcata – 601 binding (substrate nucleosome) – atgcatgcatatcattcgatctgagctcca –3' (after DraIII digestion, assembly of substrate/allosteric nucleosome and ligation to dinucleosomes).

*X.l.* symmetrical unmodified dinucleosomes used for the HMTase assays with the PRC2$^{CXC}$ mutants were obtained by reconstituting octamers with a 377 bp DNA containing two 601 sequences connected by a 35 bp linker DNA. A vector containing the 377 bp sequence was ordered from Invitrogen GeneArt and was used for PCR resulting in:

5'–atatctcgggcttatgtgatggac – 601 binding (substrate nucleosome 1) – agcgatctcaacgagtgatgctcgatactgtcata – 601 binding (substrate nucleosome 2) – gtattgaacagcgactcgggatat–3'.

The PCR products were purified as described above. Optimized ratios of octamer: DNA (usually ranging between 1.8–2.3: 1) were mixed and nucleosomes were reconstituted by gradient and stepwise dialysis against low salt buffers to a final buffer containing 25 mM Hepes, pH 7.8, 60 mM NaCl, 2 mM DTT.

## Cryo-EM data acquisition

Complexes of PHF1-PRC2 and asymmetrically modified 35 bp dinucleosomes were assembled and grids were prepared as described previously, with the difference of using 0.005% NP40 instead of 0.01% (*Poepsel et al., 2018*). Cryo-EM data were collected on an FEI Titan Krios microscope operated at 300 kV and equipped with a post-column GIF and a K2 Summit direct detector (Gatan) operated in counting mode. A total of 3467 movies were collected at a nominal magnification of 81,000x (1.746 Å/pixel) at the specimen level using a total exposure of 53 e$^-$/ Å$^2$ distributed over 60 frames and a target defocus range from 1.5 to 3 µm. Data acquisition was carried out with SerialEM.

## Cryo-EM data processing

Movies were aligned and corrected for beam-induced motion as well as dose compensated using MotionCor2 (*Zheng et al., 2017*). CTF estimation of the summed micrographs was performed with Gctf (*Zhang, 2016*) and particles were picked in Gautomatch (http://www.mrc-lmb.cam.ac.uk/kzhang/ K. Zhang, MRC LMB, Cambridge, UK) using templates created from the AEBP2-PRC2-dinucleosome cryo-EM structure (EMD-7306, *Poepsel et al., 2018*). All subsequent image processing steps were performed in Relion 3.0 (*Zivanov et al., 2018*) as shown in *Figure 1—figure supplement 2*. A total of 1,028,229 candidate particles were subjected to two rounds of initial 3D classification against a reference map (AEBP2-PRC2-dinucleosome low-pass filtered to 60 Å) and the Bayesian fudge factor (T value) set to 8. 330,482 remaining particles were subjected to two more rounds of 3D classification, this time using the best 3D model from the previous run as reference. Finally, the two best 3D models were 3D refined and further classified into 10 classes without translational and rotational sampling, using a T value of 4. From this run, the best 3D classes with the highest nominal overall resolution and rotational and translational accuracies were subjected to iterative rounds of 3D refinement, this time applying a soft mask for solvent flattening, per particle CTF refinement and Bayesian polishing. The highest nominal resolution was only achieved by combining several classes from the previous 3D run, likely due to missing particle views in one or the other individual class. The final map after postprocessing had an overall nominal resolution of 5.2 Å, as determined from the gold-standard FSC criterion of 0.143 (*Rosenthal and Henderson, 2003*; *Figure 1—figure supplement 1D*). The density (Overall PHF1-PRC2:di-Nuc) with fitted models is shown in *Figure 1C* and in *Figure 1—figure supplement 1E* using UCSF ChimeraX (*Goddard et al., 2018*). Local resolution estimation was performed in Relion 3.0 and is shown in *Figure 1—figure supplement 1B*. The spherical angular distribution of all particles in the final model is shown in *Figure 1—figure supplement 1C*.

To further improve the resolution and map details of the region around the H3 N-terminus, particle subtraction and focused 3D refinement was applied (*Bai et al., 2015*; *Zhou et al., 2015*; *Ilca et al., 2015*). Using a mask generated with UCSF Chimera (*Pettersen et al., 2004*) and Relion

3.0 the signal of the allosteric nucleosome as well as parts of PRC2 (EED and EZH2$_{allo}$) was subtracted from all particle images. These signal subtracted particles were then subjected to focused 3D refinement using a soft mask around the substrate nucleosome and EZH2$_{sub}$. This yielded a 4.4 Å map (EZH2$_{sub}$-Nuc$_{sub}$) as determined from the gold-standard FSC criterion of 0.143 (*Rosenthal and Henderson, 2003*; *Figure 1—figure supplement 3B*). Local resolution estimation is shown in *Figure 1—figure supplement 3A*. For model building and depiction, the final density was further sharpened (applied b – factor: - 66) using the Multisharpen function in Coot (*Emsley et al., 2010*) (e.g. in *Figure 1F*, *Figure 1—figure supplement 3D*).

To confirm the side-chain information visible in the Coot sharpened map, Phenix Resolve density modification was run on the two half maps generated from the 3D refinement of the EZH2$_{sub}$-Nuc$_{sub}$ map (*Terwilliger et al., 2020*). The resolution of the map according to Phenix cryo-EM density modification output improved to 4 Å and the resulting map was used as an additional guideline for model building as well as for depiction (in *Figure 1—figure supplement 4A-D*).

## Cryo-EM data fitting, modeling and refinement

Available crystal structures were fitted into the final maps using rigid-body fitting in UCSF Chimera and all manual remodeling, morphing and building was performed in Coot. For PRC2, the crystal structure of the catalytic lobe of human PRC2 (PDB: 5HYN *Justin et al., 2016*; and comparing the fitted model to the cryo EM model of AEBP2-JARID2-PRC2 PDBs: 6C23 and 6C24 *Kasinath et al., 2018*) was used. Since the SBD helix and the SANT1 helix bundle of the crystal structure was not accommodated well by the corresponding EM density, this region was fitted separately. A model of a dinucleosome with linker DNA (Supplementary dataset one in *Poepsel et al., 2018*, including crystal structures of nucleosomes, PDB 3LZ1, *Vasudevan et al., 2010*, also PDB 1AOI, *Luger et al., 1997*, also PDB 6T9L, *Wang et al., 2020*, was fitted.

The above described overall model was then used as a starting model for fitting and building EZH2$_{sub}$-Nuc$_{sub}$ into the focused map. Where possible, missing parts in the model were built de-novo, that is the H3 N-terminal tail (residues 30–37) between the catalytic site of PRC2 and the substrate histone. Available information from crystal/cryo EM structures was used as a guide (PRC2 with H3 peptide bound: PDB: 5HYN *Justin et al., 2016* and cryo EM model of AEBP2-JARID2-PRC2 PDBs: 6C23 and 6C24 *Kasinath et al., 2018*), and high-resolution structures of nucleosomes (PDB 1AOI and PDB 6T9L) (*Luger et al., 1997*; *Wang et al., 2020*). Parts of EZH2$_{sub}$-Nuc$_{sub}$ model were then fitted using the morph fit routine in Coot or manually (*Casañal et al., 2020*). Secondary structure restraints for real-space refinement were generated automatically with phenix.secondary_structure_restraints (*Sobolev et al., 2015*) and manually curated. Hydrogens were added and the model was real-space refined with a resolution- cutoff of 4.4 Å with Phenix (*Afonine et al., 2018*) (phenix-1.18rc1-3777), using reference structures (PDB 6T9L, *Wang et al., 2020*, and PDB 1AOI, *Luger et al., 1997* for nucleosome and one copy of the human PRC2 crystal structure generated from PDB 5HYN (*Justin et al., 2016* ), applying strict secondary structure and Ramachandran restraints.

Our final model includes the modeled side chains of the fitted crystal/cryo-EM structures. This is in our opinion supported by the data as the substrate nucleosome protein core is resolved to app. 4 Å (*Figure 1—figure supplement 3A*) and the map in these regions shows clear bulky side-chain information (*Figure 1—figure supplement 3D*). The EZH2 density is of worse quality however even at lower resolution side chains likely contribute to the signal in the particle images and thereby an overall good model to map fit (in our case given by the high CC values as well as FSC$_{modelvsmap}$) is arguably only ensured in the presence of side chains. However we caution readers against in interpreting our model at side-chain resolution in poorly resolved regions.

Structures were visualized with UCSF ChimeraX (*Goddard et al., 2018*) and PyMOL2 (https://pymol.org/2/).

## Electrophoretic mobility shift assay (EMSA)

EMSAs on a 1.2% agarose gel in 0.4x TBE Buffer with 45 nM 6-FAM - labeled mononucleosomes (unmodified wt *X.l.* for bandshifts with the PRC2$^{CXC}$ mutants, unmodified wt *D.m.* and *D.m* H3Kc36me3 *Simon et al., 2007*) trimethyllysine analog containing nucleosomes) and increasing PRC2 concentrations (concentrations indicated in the figures above the gels) were performed in

triplicates as described in *Choi et al., 2017*. A Typhoon FLA 9500 scanner and the Fiji software was used for densitometric analysis of the 6-FAM signal (*Schindelin et al., 2012*). Background correction and calculation of the fractions of bound nucleosomes was performed with R using tidyverse (https://www.r-project.org/). In detail: two parts were boxed out in each lane: 1. unbound nucleosomes ('unbound' box) and 2. shifted nucleosomes ('bound', everything above 'unbound'). The boxed-out signals were integrated and background corrected by subtracting the respective control ('bound' background of lane one for 'bound' boxes and 'unbound' background of lane 10 for 'unbound' boxes). To calculate the fraction of bound vs. unbound nucleosomes, the value for 'bound' nucleosome in each lane was divided by the total signal (sum of bound and unbound) of the same lane. Hill function fitting and illustration of the plot were subsequently performed with Prism 8 (GraphPad).

## Histone methyltransferase (HMTase) assay

For all HMTase assays, 446 nM of mononucleosomes or 223 nM of dinucleosomes were incubated with indicated amounts of the different PRC2 complexes, in a reaction buffer containing 20 mM HEPES pH 7.8, 50 mM NaCl, 2.5 mM $MgCl_2$, 5% glycerol, 0.25 mM EDTA, 0.5 mM DTT and 80 µM S-adenosylmethionine (SAM). Reactions were allowed to proceed for 90 min at RT before quenching by the addition of 1x (final concentration) SDS loading buffer and heat inactivation at 95°C for 5 min. Proteins were separated by electrophoresis on a 16% (w/v) SDS gel, transferred to a nitrocellulose membrane and probed with antibodies against H3K27me3 (Millipore, 07–449), H3K27me1 (Millipore, 07–448) and H4 (Abcam, ab10158). For quantification, HMTase reactions and the corresponding western blots on *D.m.* unmodified, H3Kc36me3, $H3^{K36A/R}$ mononucleosomes were performed in triplicates and subjected to densitometric analysis (Chemiluminescence signal, ImageQuant LAS 4000). The integrated densitometric signal (band) in each lane was background corrected against the control lane (lane 1, no PRC2 in the reaction) and normalized with respect to the lane containing the highest amount (i.e. 100%) of PRC2 on unmodified nucleosomes (lane 4). The relative amounts of tri-methylation/monomethylation for all other lanes were calculated with respect to lane 4. Graphical representations were made with Prism 8 (GraphPad).

## Mass spectrometry (MS)

500 nM of PRC2 or $PRC2^{CXC>A}$ were incubated with 2 µM of either unmodified or $H3_{18-42}$ peptide containing the K36me3 modification in HMTase reaction buffer (described above) and methyltransferase activity was allowed to proceed over night at RT. Reactions were then quenched with 1% trifluoroacetic acid (TFA). Home-made stage tips with poly(styrenedivinylbenzene) copolymer (SDB-XC) were used to remove PRC2 from the reactions (*Rappsilber et al., 2007*). First, stage tips were washed with methanol, followed by a second wash with buffer B (0.1% (v/v) formic acid, 80% (v/v) acetonitrile). The SDB-XC material was then equilibrated with buffer A (0.1% (v/v) formic acid) and 40 µl of sample was applied and washed several times. Finally, samples were eluted using buffer B and introduced into the Bruker maXis II ETD mass spectrometer by flow injection of 20 µl sample using an Agilent HPLC at a flow rate of 250 µl/min and 0.05% TFA in 70% acetonitril:H2O as solvent for ESI-MS time-of-flight analysis. Peptides were ionized at a capillary voltage of 4500 V and an end plate offset of 500 V. Full scan MS spectra (200–1600 m/z) were acquired at a spectra rate of 1 Hz and a collision cell energy of 15 eV.

Raw data files were processed using Bruker Compass DataAnalysis. The m/z spectra were deconvoluted (maximum entropy method) with an instrument resolving power of 10,000 and the resulting neutral spectra peaks were integrated. For quantification, the experiment was performed in triplicates. The sum of the monomethylation peak areas was divided by the sum of the first 4 peaks of the input peptide together with the sum of the monomethylation peak areas. Illustration of the quantification was subsequently performed with Prism 8 (GraphPad). A Welch's t-test was calculated to show the nonsignificant difference between the activity of PRC2 on unmodified or H3K36me3 peptide.

## Construction of histone transgenes to generate $H3^{K36A}$ and $H3^{K36R}$ strains

Site directed mutagenesis on *pENTR221-HisGU.WT*, *pENTRL4R1-HisGU.WT* and *pENTRR2L3-HisGU.WT* (*Günesdogan et al., 2010*) was used to mutate histone H3K36 to alanine or arginine. The final constructs *pfC31-attB-3xHisGU.H3K36A* and *pfC31-attB-3xHisGU.H3K36R* were generated by Gateway LR recombination of above vectors and integrated at attP sites VK33 (BDSC 9750) and 86Fb (BDSC 130437). The full genotypes of animals used in the study are described below.

### *Drosophila* strains and genotypes

The following strains were used in this study:

Oregon-R
$w; Df(2L)His^C FRT40A/Df(2L)His^C FRT40A; 12xHisGU^{wt}/12 xHisGU^{wt}$ (*McKay et al., 2015*)
$w; Df(2L)His^C FRT40A/CyO ubi-GFP; 12xHisGU^{H3K36R}/TM6B$ (*McKay et al., 2015*)
$w; Df(2L)His^C FRT40A/CyO twi:Gal4 UAS:GFP; 3xHisGU^{H3K36A}(VK33) 3xHisGU^{H3K36A}(86Fb)/ 3xHisGU^{H3K36A}(VK33) 3xHisGU^{H3K36A}(86Fb)$ (generated in this study)
$w; Df(2L)His^C FRT40A/CyO twi:Gal4 UAS:GFP; 3xHisGU^{H3K36R}(VK33) 3xHisGU^{H3K36R}(86Fb)/ 3xHisGU^{H3K36R}(VK33) .3xHisGU^{H3K36R}(86Fb)$ (generated in this study)
$w; Df(2L)His^C FRT40A/CyO ubi:GFP; 3xHisGU^{H3K27R}(68E) 3xHisGU^{H3K27R} (86Fb)/3xHisGU^{H3K27R} (68E) 3xHisGU^{H3K27R} (86Fb)$ (*Pengelly et al., 2013*)
$w hs-flp; w; hs-nGFP FRT40A/hs nGFP FRT40; 3xHisGU^{H3K27R}(68E) 3xHisGU^{H3K27R}(86Fb)/3xHis-GU^{H3K27R}(68E) 3xHisGU^{H3K27R} (86Fb)$ (*Pengelly et al., 2013*)
$w hs-flp; M(2)25A ubi-GFP FRT40A/CyO yw; esc^6 b pr/CyO, P[esc^+]$
$In(2LR) Gla/CyO, esc^2$
$w hs-flp; hs-nGFP FRT2A/hs nGFP FRT2A$

The following genotypes were used for the experiments shown in:

*Figure 4*
wt: $Df(2L) HisC FRT40/Df(2L) HisC FRT40; 12xHisGU^{wt}(VK33)/12xHisGU^{wt}(VK33)$
$H3^{K36R}$: $Df(2L) HisC FRT40/Df(2L) HisC FRT40; 12xHisGU^{H3K36R}(VK33)/TM6B$
$H3^{K36R}$: $w; Df(2L)His^C FRT40A/Df(2L)His^C FRT40A; 3xHisGU^{H3K36R}(VK33) 3xHisGU^{H3K36R}(86Fb)/ 3xHisGU^{H3K36R}(VK33) 3xHisGU^{H3K36R}(86Fb)$
$H3^{K36A}$: $Df(2L)His^C FRT40A/Df(2L)His^C FRT40A; 3xHisGU^{H3K36A}(VK33) 3xHisGU^{H3K36A}(86Fb)/ 3xHisGU^{H3K36A}(VK33) 3xHisGU^{H3K36A}(86Fb)$

*Figure 5A, C, E*
wt: $Df(2L) HisC FRT40/Df(2L) HisC FRT40; 12xHisGU^{wt}(VK33)/12xHisGU^{wt}(VK33)$
$H3^{K36R}$: $Df(2L) HisC FRT40/Df(2L) HisC FRT40; 12xHisGU^{H3K36R}(VK33)/TM6B$

*Figure 5B, D, F*
wt: Oregon-R
$H3^{K36A}$: $w; Df(2L)His^C FRT40A/Df(2L)His^C FRT40A; 3xHisGU^{H3K36A}(VK33) 3xHisGU^{H3K36A}(86Fb)/ 3xHisGU^{H3K36A}(VK33) 3xHisGU^{H3K36A}(86Fb)$
$H3^{K36R}$: $w; Df(2L)His^C FRT40A/Df(2L)His^C FRT40A; 3xHisGU^{H3K36R}(VK33) 3xHisGU^{H3K36R}(86Fb)/ 3xHisGU^{H3K36R}(VK33). 3xHisGU^{H3K36R}(86Fb)$

*Figure 6A*
wt: $Df(2L) HisC FRT40/Df(2L) HisC FRT40; 12xHisGU^{wt}(VK33)/12xHisGU^{wt}(VK33)$
$H3^{K36R}$: $Df(2L) HisC FRT40/Df(2L) HisC FRT40; 12xHisGU^{H3K36R}(VK33)/TM6B$
$H3^{K36R}$: $w; Df(2L)His^C FRT40A/Df(2L)His^C FRT40A; 3xHisGU^{H3K36R}(VK33) 3xHisGU^{H3K36R}(86Fb)/ 3xHisGU^{H3K36R}(VK33) 3xHisGU^{H3K36R}(86Fb)$.
$H3^{K36A}$: $w; Df(2L)His^C FRT40A/Df(2L)His^C FRT40A; 3xHisGU^{H3K36A}(VK33) 3xHisGU^{H3K36A}(86Fb)/ 3xHisGU^{H3K36A}(VK33) 3xHisGU^{H3K36A}(86Fb)$.
$H3^{K27R}$: $w; Df(2L)His^C FRT40A/Df(2L)His^C FRT40A; 3xHisGU^{H3K27R}(68E) 3xHisGU^{H3K27R} (86Fb)/ 3xHisGU^{H3K27R} (68E) 3xHisGU^{H3K27R} (86Fb)$
$esc^-$: $esc^6 b pr/CyO, esc^2$ ($esc^{mat- zyg-}$ obtained as progeny from $esc^6 b pr/CyO, esc^2$ parents).

*Figure 6B*
wt: $Df(2L) HisC FRT40/Df(2L) HisC FRT40; 12xHisGU^{wt}(VK33)/12xHisGU^{wt}(VK33)$

*H3$^{K36R}$: Df(2L) HisC FRT40/Df(2L) HisC FRT40; 12xHisGU$^{H3K36R}$(VK33)/TM6B*

*Figure 6C*
*wt: Df(2L) HisC FRT40/Df(2L) HisC FRT40; 12xHisGU$^{wt}$(VK33)/12xHisGU$^{wt}$(VK33)*
*H3$^{K36R}$: Df(2L) HisC FRT40/Df(2L) HisC FRT40; 12xHisGU$^{H3K36R}$(VK33)/TM6B*
*H3$^{K27R}$: w hs-flp; Df(2L)His$^C$ FRT40A/hs-nGFP FRT40A; 3xHisGU$^{H3K27R}$(68E)3xHis-GU$^{H3K27R}$(86Fb)/3xHisGU$^{H3K27R}$(68E)3xHisGU$^{H3K27R}$(86Fb)*

*Figure 6D*
*wt: Df(2L) HisC FRT40/Df(2L) HisC FRT40; 12xHisGU$^{wt}$(VK33)/12xHisGU$^{wt}$(VK33)*
*H3$^{K36R}$: Df(2L) HisC FRT40/Df(2L) HisC FRT40; 12xHisGU$^{H3K36R}$(VK33)/TM6B*
*H3$^{K36A}$: w hs-flp; Df(2L)HisC FRT40A/M(2)25AubiGFP FRT40; 3xHisGU$^{H3K36A}$(VK33) 3xHis-GU$^{H3K36R}$(86Fb)/ +*
*H3$^{K27R}$: w hs-flp; Df(2L)HisC FRT40A/M(2)25A ubi-GFP FRT40; 3xHisGU$^{H3K27R}$(68E) 3xHis-GU$^{H3K27R}$ (86Fb)/ +*

*Figure 6—figure supplement 1*
*wt: w hs-flp; hs–nGFP FRT2A/hs nGFP FRT2A*
*H3$^{K36R}$: Df(2L) HisC FRT40/Df(2L) HisC FRT40; 12xHisGU$^{H3K36R}$(VK33)/TM6B*
*H3$^{K27R}$: w hs-flp; Df(2L)His$^C$ FRT40A/hs-nGFP FRT40A; 3xHisGU$^{H3K27R}$(68E)3xHis-GU$^{H3K27R}$(86Fb)/3xHisGU$^{H3K27R}$(68E)3xHisGU$^{H3K27R}$(86Fb)*

## Comparison of the lethality of H3$^{K36R}$ and H3$^{K36A}$ mutants

The difference in the lethality phase of the H3$^{K36R}$ mutants generated in this study compared to H3$^{K36R}$ mutants in the strain from Matera and colleagues was unexpected because both strains the Df(2L)HisC homozygotes carry 12 copies of the HisGU$^{H3K36R}$ cassette (i.e. four 3x HisGU$^{H3K36R}$ arrays in our strain and a single 12x HisGU$^{H3K36R}$ array in the strain from Matera and colleagues). A possible explanation for the poorer survival of H3$^{K36R}$ mutants in the strain generated here could be that histone transgene expression from the 3xHisGU$^{H3K36R}$ miniarrays is for some reason less effective that in the case of the 12x HisGU$^{H3K36R}$ array. We also note that a recent study reported that among Df(2L)HisC homozygotes that carry 20 HisGU$^{H3K36A}$ copies, about 50% of the mutant animals develop up to the pupal stages (Zhang et al., 2019). Zhang et al have not analyzed their H3$^{K36A}$ mutants any further but it is possible that the higher copy number of the HisGU$^{H3K36A}$ cassette accounts for the better survival compared to the H3$^{K36A}$ strain generated in this study. Further studies will be needed to explore whether H3$^{K36R}$ and H3$^{K36A}$ mutants show comparable phenotypes in larvae.

## Immunohistochemistry and immunofluorescence stainings

Embryos of the appropriate genotypes listed above were identified by the lack of GFP marked balancer chromosomes, fixed and stained with Abd-B antibody, following standard protocols. Imaginal discs from third instar larvae were stained with Antp and Cy3-labeled secondary antibodies following standard protocols. For clonal analysis (Figure 3D), clones were induced 96 hr before analyses by heat-shocked induced expression of Flp recombinase in the genotypes listed above.

## ChIP-seq analysis in *Drosophila* embryos and in larval tissues

### Embryo collection, chromatin preparation, and ChIP

21-24 hr old *wt, H3$^{K36A}$* embryos (see above for details of genotypes) were dechorionated, quick-frozen in liquid N2 and stored at -80˚C. 5 µl of thawed embryos were homogenized in 5 mL of fixing solution (60 mM KCl, 15 mM NaCl, 4 mM MgCl$_2$, 15 mM Hepes pH 7.6, 0.5% Triton X-100, 0.5 mM DTT, protease inhibitors, 0.9% Formaldehyde) at r.t. The homogenate was filtered through a strainer (Greiner Bio-One, EASYstrainer 100 µm, #542 000) and incubated for 10 min with frequent gentle shaking. Cross-linking was stopped by the addition of 450 µl of 2.5 M Glycine. Fixed nuclei were washed with 1 ml of buffer A1 (60 mM KCl, 15 mM NaCl, 4 mM MgCl$_2$, 15 mM Hepes pH 7.6, 0.5% Triton X-100, 0.5 mM DTT, protease inhibitors), washed with 1 ml of pre-lysis buffer (140 mM NaCl, 15 mM Hepes pH 7.6, 1 mM EDTA, 0.5 mM EGTA, 1% Triton X-100, 0.5 mM DTT, 0.1% Na Deoxycholate, protease inhibitors), resuspended in 1 ml of lysis buffer (140 mM NaCl, 15 mM Hepes pH 7.6, 1 mM EDTA, 0.5 mM EGTA, 1% Triton X-100, 0.5 mM DTT, 0.1% Na Deoxycholate, protease

inhibitors, 0.1% SDS, 0.5% N-laurylsarcosine), incubated at least 10 min at 4°C with shaking, and transferred into milliTUBES 1 ml AFA Fiber (100) (Covaris, #520130) for sonication. Sonication was performed in a Covaris S220 AFA instrument using the following setup: 140W (peak incident power) / 5% (duty cycle) / 200 (cycle per burst) / 15 min. Insoluble material was removed by centrifugation in an Eppendorf centrifuge at 14000 rpm (10 min at 4°C). Input chromatin was quantified by measuring DNA concentration after decrosslinking using Qubit (Thermo Scientific) and 250 ng of chromatin were used for each ChIP experiment. 250 ng of an independently prepared batch of *D. pseudoobscura* chromatin were spiked-in in each ChIP experiment for subsequent normalization of the ChIP-seq datasets. The rest of the ChIP protocol was performed as described in *Bonnet et al., 2019*. For each condition, the ChIP experiment was performed in duplicates from two biologically independent chromatins. ChIP on hand-dissected CNS and imaginal disc tissues from 3$^{rd}$ instar *wt* or *H3$^{K36R}$* homozygous larvae (see above for details on genotypes) was performed as described *Laprell et al., 2017* with the difference *D. pseudoobscura* chromatin was spiked in at a 1:1 ratio of dm / dp chromatin.

## Library preparation and sequencing

Library preparation for sequencing was performed with TruSeq kits from Illumina. Illumina systems (NextSeq 500) were used for paired-end DNA sequencing. All reads were aligned using STAR (*Dobin et al., 2013*) to the *D. melanogaster* dm6 genome assembly (*dos Santos et al., 2015*) and to the *D. pseudoobscura* dp3 genome assembly (Nov. 2004, FlyBase Release 1.03). Only sequences that mapped uniquely to the genome with a maximum of two mismatches were considered for further analyses.

## Identification of H3K36me2 and H3K27me3 enriched regions

The Bioconductor STAN-package (*Zacher et al., 2017*) was used to define the location of H3K36me2-enriched regions. The seven chromosome arms (X, 2L, 2R, 3L, 3R, 4 and Y) defined in the dm6 genome assembly were segmented in 200 bp bins. STAN annotated each of these bins into 1 of 3 'genomic states' based on the number of H3K36me2 ChIP-seq reads and the number of input reads overlapping with each bin in *wildtype* embryos or larvae. These 3 'genomic states' corresponded to: 'H3K36me2 enriched' regions; 'low or no H3K36me2' regions and 'no input' regions. The Poisson Lognormal distribution was selected and fitting of hidden Markov models was performed with a maximum number of 100 iterations. Stretches of consecutive bins annotated as 'H3K36me2 enriched' regions were sometimes separated by a few bins showing another type of annotation (i.e. 'no input'). To define a relevant set of H3K36me2 enriched regions, we considered that if stretches of consecutive bins annotated as 'H3K36me2 enriched' regions are not separated by more than 7 Kb, they can be fused. High-level H3K27me3 domains previously defined using the same Bioconductor STAN-package in *Bonnet et al., 2019* were used in this study.

## Normalization and visualisation of H3K27me3 and H3K36me2 ChIP-Seq datasets

The proportion of *D. pseudoobscura* reads as compared to *D. melanogaster* reads in input and in samples was used to normalize the H3K36me2 and H3K27me3 ChIP-seq datasets from *H3$^{K36A}$* and *H3$^{K36R}$* mutants to the corresponding *wildtype* H3K36me2 and H3K27me3 ChIP-seq datasets respectively (see *Supplementary file 2*). Chip-seq tracks shown in *Figure 4* show the average of the two replicates that were performed for each condition. Y-axes of ChIP-seq tracks correspond to normalized numbers of mapped reads per million reads per 200 bp bin.

## Calculation of read coverage

In *wildtype* and *H3$^{K36A}$* and *H3$^{K36R}$* mutant conditions, H3K36me2 and H3K27me3 ChIP-seq read coverages across gene bodies were computed on genomic intervals starting 750 bp upstream transcription start sites and ending 750 bp downstream transcription termination sites. Read coverage is defined as the normalized number of mapped reads per million reads from a ChIP-seq dataset divided by the number of mapped reads per million reads from the corresponding input dataset across a genomic region. Among the *D. melanogaster* Refseq genes, approximately 10800 and 9200 are overlapping with H3K36me2 enriched regions, approximately 1030 and 1030 genes are

overlapping with high-level H3K27me3 domains and 5400 and 6300 are localized in other genomic regions in embryos and larvae, respectively.

## *Drosophila* nuclear and cell extracts for western blot analysis

For embryonic total nuclear extracts, nuclei from 21 to 24 hr old *wt*, $H3^{K36A}$ or $H3^{K36A}$ mutant embryos were purified and quantified as described in *Bonnet et al., 2019*. Pellets of nuclei were resuspended in appropriate volumes of SDS sample buffer proportional to the number of nuclei in each pellet. Extracts were then sonicated in a Bioruptor instrument (Diagenode) (eight cycles (30 s ON/30 s OFF), high power mode), incubated at 75 °C for 5 min and insoluble material was removed by centrifugation at 14000 rpm for one mn at r.t.

Total cell extracts from imaginal disc tissues were prepared by resuspending hand-dissected disc tissues in SDS sample buffer. Extracts were then sonicated, incubated at 75 °C for 5 min and insoluble material was removed by centrifugation.

### Antibodies

**For ChIP analysis:**

| | |
|---|---|
| Rabbit monoclonal anti-H3K27me3 | Cell Signaling Technology #9733 |
| Rabbit polyclonal anti-H3K36me2 | Abcam #9049 |

**For western blot analysis on embryonic and larval extracts:**

| | |
|---|---|
| Rabbit monoclonal anti-H3K27me3 | Cell Signaling Technology #9733 |
| Rabbit polyclonal anti- H3K27me3 | Millipore #07-449 |
| Rabbit polyclonal anti-H3K27me1 | Millipore #07-448 |
| Rabbit monoclonal anti-H3K36me3 | Cell Signaling Technology #4909 |
| Rabbit monoclonal anti-H3K36me2 | Cell Signaling Technology #2901 |
| Rabbit polyclonal anti-H2B | (against full-length recombinant D.m. H2B) |
| Rabbit polyclonal anti-H4 | Abcam #10158 |
| Rabbit polyclonal anti-Caf1 | *Gambetta et al., 2009* |

**For immunohistochemistry and immunofluorescence analysis:**

| | |
|---|---|
| Mouse monoclonal anti-Abd-B | DSHB (1A2E9) |
| Mouse monoclonal anti-Antp | DSHB (8C11) |
| Rabbit monoclonal anti-H3K27me3 | Cell Signaling Technology #9733 |

## Acknowledgements

We thank Eva Nogales for generous advice, sharing of expertise and for hosting KF for grid preparation. We thank Tom Cech for stimulating discussions. We thank JR Prabu for excellent computing support, S Uebel, E.Weyher, R Kim and A Yeroslaviz of the MPIB core facilities and Martin Spitaler and Giovannie Cardone of the MPIB imaging facility for excellent technical support and S Schkoelziger and S Schmähling for help with some of the experiments. This work was supported by the Deutsche Forschungsgemeinschaft (SFB1064) and the MPG. ChIP-seq data have been deposited in GEO (accession number: GSE148254). The structural data are deposited in the EMDB (EMD-11910 and EMD-11912) and PDB (7AT8).

## Additional information

### Funding

| Funder | Grant reference number | Author |
|---|---|---|
| Deutsche Forschungsgemeinschaft | SFB1064 | Jürg Müller |
| Max-Planck-Gesellschaft | | Jürg Müller |

The funders had no role in study design, data collection and interpretation, or the decision to submit the work for publication.

## Author contributions
Ksenia Finogenova, Jacques Bonnet, Conceptualization, Data curation, Formal analysis, Validation, Investigation, Visualization, Methodology, Writing - review and editing; Simon Poepsel, Katja Finkl, Katharina Schmid, Claudia Litz, Mike Strauss, Methodology; Ingmar B Schäfer, Christian Benda, Validation, Methodology; Jürg Müller, Conceptualization, Data curation, Formal analysis, Supervision, Funding acquisition, Validation, Investigation, Visualization, Methodology, Writing - original draft, Writing - review and editing

## Author ORCIDs
Ksenia Finogenova (iD) https://orcid.org/0000-0002-7512-8637
Jürg Müller (iD) https://orcid.org/0000-0003-2391-4641

## Decision letter and Author response
Decision letter https://doi.org/10.7554/eLife.61964.sa1
Author response https://doi.org/10.7554/eLife.61964.sa2

# Additional files

## Supplementary files
• Source code 1. PDB model of PHF1-PRC2:di-Nuc complex (related to *Figure 1*) with the corresponding description text file. A pseudoatomic pdb model of the overall PHF1-PRC2:di-Nuc structure. Available crystal structures were fitted into the final map: for PRC2, the crystal structure of the catalytic lobe of human PRC2 (PDB: 5HYN, *Justin et al., 2016*) was fitted. A model of a dinucleosome with linker DNA (Supplementary dataset one in *Poepsel et al., 2018*), including crystal structures of nucleosomes, (PDB 3LZ1, *Vasudevan et al., 2010* , also PDB 1AOI, *Luger et al., 1997*, and PDB 6T9L, *Wang et al., 2020*) was used. We caution the reader that this model did not undergo real-space-refinement. PRC2: Chain A: EZH2 Chain B: EED Chain C: SUZ12 Chain D – Chain K = substrate nucleosome histones Chain D: Histone H3 #1 Chain E: Histone H4 #1 Chain F = Histone H2A #1 Chain G = Histone H2B #1 Chain H: Histone H3 #2 Chain I: Histone H4 #2 Chain J = Histone H2A #2 Chain K = Histone H2B #2 Chain L – Chain S = allosteric nucleosome histones Chain L: Histone H3 #1 Chain M: Histone H4 #1 Chain N = Histone H2A #1 Chain O = Histone H2B #1 Chain P: Histone H3 #2 Chain Q: Histone H4 #2 Chain R = Histone H2A #2 Chain S = Histone H2B #2 Chain T and chain U = nucleosomal DNA Widom601+linker DNA

• Supplementary file 1. Cryo-electron microscopy data collection summary, processing statistics and model.

• Supplementary file 2. Number of aligned reads to the *D. melanogaster* and *D. pseudoobscura* genomes from ChIP and input datasets and normalization process (related to *Figure 5* and *Figure 5—figure supplement 1*).

• Transparent reporting form

## Data availability
The sequence datasets generated in this study have been deposited in GEO (accession number: GSE148254). The protein structure data reported in this study have been deposited in PDB under the accession code 7AT8 and in the EMDB under the accession codes EMD-11910 and EMD-11912.

The following datasets were generated:

| Author(s) | Year | Dataset title | Dataset URL | Database and Identifier |
|---|---|---|---|---|
| Müller J, Bonnet J | 2020 | Inhibition of PRC2 activity by H3K36 methylation | https://www.ncbi.nlm.nih.gov/geo/query/acc.cgi?acc=GSE148254 | NCBI Gene Expression Omnibus, GSE148254 |

| Finogenova K, Benda C, Schäfer IB, Poepsel S, Strauss M, Müller J | 2020 | Histone H3 recognition by nucleosome-bound PRC2 subunit EZH2 | https://www.rcsb.org/structure/7AT8 | RCSB Protein Data Bank, 7AT8 |
|---|---|---|---|---|
| Finogenova K, Benda C, Schäfer IB, Poepsel S, Strauss M, Müller J | 2020 | Histone H3 recognition by nucleosome-bound PRC2 subunit EZH2 | https://www.emdatare-source.org/EMD-11910 | Electron Microscopy Data Bank, EMD-11910 |
| Finogenova K, Benda C, Schäfer IB, Poepsel S, Strauss M, Müller J | 2020 | Cryo-EM map of PHF1-PRC2 on a heterodimeric dinucleosome | https://www.emdatare-source.org/EMD-11912 | EMDataBank, EMD-11912 |

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
