## [Decision Letter]

**Acceptance summary:**

This manuscript provides a structural explanation for how histone H3K36 methylation, an active transcription mark, prevents H3K27 methylation by Polycomb complexes which repress gene expression. These results provide substantial insights into crosstalk among histone modifications.

**Decision letter after peer review:**

Thank you for submitting your article "Structural basis for PRC2 decoding of active histone methylation marks H3K36me2/3" for consideration by *eLife*. Your article has been reviewed by three peer reviewers, including Jerry L Workman as the Reviewing Editor and Reviewer #1, and the evaluation has been overseen by Cynthia Wolberger as the Senior Editor.

The reviewers have discussed the reviews with one another and the Reviewing Editor has drafted this decision to help you prepare a revised submission.

Summary:

In this manuscript, the authors determine cryo-EM structures of human PRC2 on dinucleosomes. Several such structures have been published in recent years. The novel aspects presented in this study, are focused on DNA-interacting interphases of EZH2 and EED as well as a cis-acting role of unmodified H3K36 in facilitating efficient H3K27 methylation by PRC2.

H3K36me2/3 has been proposed to influence PRC2 function in several ways: H3K36me3 has been shown to inhibit PRC2 catalytic activity in vitro, but has also been suggested as a potential recruitment mechanism to transcribed chromatin via recognition by the Tudor domains of PCL proteins. Here, the authors demonstrate, through in vitro and in vivo approaches, that the substitution of H3K36 with H3K36R or H3K36A, and therefore the absence of an unmodified H3K36, leads to global reduction of H3K27me3. Moreover, they show that PRC2 binds to mononucleosomes with H3K36me3, however, that they are poor PRC2 substrates. Taken together with the presented cryo-EM data, the authors provide a plausible mechanism for the inhibitory effects of H3K36me2/me3 on PRC2 activity.

Overall, this is a well-performed study, which addresses an open question in the field. The manuscript is well-written and the conclusions are generally well-supported – however, some additional experiments and alternative interpretations should be taken into consideration and incorporated in the manuscript (see specific comments below).

Essential revisions:

1) One of the primary conclusions reached by the authors is that the cleft which H3K36 binds at the EZH2-nucleosomal DNA interface does not fit H3K36A and H3K36R mutations or methylated forms of H3K36 (i.e., H3K36me2 and H3K36me3), leading to allosteric inhibition of PRC2. The structure of the nucleosome-PRC2 complex (Figure 1—figure supplement 4E) illustrates that the side chain of K36 engages in few interactions in this cleft and does not contact the nucleosomal DNA. Furthermore, there is a relatively large unoccupied space in the cleft between the K36 side chain and the DNA, suggesting that amino acids larger than lysine can fit within it. Modeling of an K36R mutation indicates that the arginine side chain can be accommodated in the K36 binding cleft without clashes. Given their similar size, methylated forms of K36 could also fit within the H3K36 binding cleft. The claim that the cleft at the EZH2-DNA interface fits lysine (K36) but cannot accommodate larger amino acids is unconvincing in the absence of additional data.

2) The mechanism by which H3K36 binding at the EZH2-nucleosomal DNA interface allosterically regulates PRC2 enzymatic activity is not clearly established. The nucleosome-PRC2 structure shows one potential polar interaction between the K36 side chain and the backbone of Gln570 in EZH2, implying a lack of specificity in recognizing K36. In contrast, other proteins that specifically recognize unmodified lysines in the histone H3 N-terminal tail have been shown to do so through a network of hydrogen bonds and/or salt bridge interactions (e.g., H3K4 recognition by the BHC80 PHD domain; Lan et al. (2007) Nature 448, 718-22). It is unlikely that a single (potential) polar interaction allosterically regulates PRC2 methyltransferase activity.

3) The authors attribute the loss of H3K27me3 in the *Drosophila* H3K36A and H3K36R mutants to allosteric inhibition of PRC2. However, these mutations also diminish H3K36me3 in developing *Drosophila* embryos and larvae. The *Drosophila* PRC2 accessory subunit Pcl has been shown to bind H3K36me3, analogous to its mammalian homolog PHF1 (Ballare et al., 2012; Musselman et al., 2012). Thus, the global loss of H3K36me3 may result in either diminished recruitment or mislocalization of PRC2 in chromatin. The authors should examine whether the H3K36A and H3K36R mutations affect PRC2 localization within chromatin relative to wild type *Drosophila*.

4) The authors use full length human PRC2 in a complex with PHF1 for the Cryo-EM studies and in nucleosome binding experiments. In the Cryo-EM studies, they are not able to detect the density for the N-terminal part of SUZ12, RBBP4 and the N-terminal part of PHF1. In several publications these parts of the PRC2 complex have been shown to be important for the localization of PRC2 to specific sites in the genome. Although, the authors do not test this directly, the N-terminal part of SUZ12 and RBBP4 are most likely important for the binding (affinities) that the authors measure in EMSA. If this indeed is correct, the authors should take this into their discussion of the relative contribution of the various parts of PRC2 to DNA and nucleosome binding affinities.

5) The description in the text states that PRC2^CXC>A^ and PRC2^EED>A^ have similar binding strength deficits in the EMSA assays, which is also shown in the quantification (Figure 2B) – however, based on the gels presented (Figure S5A), the EED mutant appears to perform similarly to the PRC2 WT? Either it is a poorly chosen example (of the three replicates used for quantification) or we would question whether the gel image is of sufficient quality for this quantification.

6) The EMSA experiments (Figure 2A and S5A) are performed solely on mononucleosomes, presumably meaning that only one of the studied PRC2 binding surfaces (CXC vs EED) can be engaged at a time. In this scenario, mutation of one binding surface presumably would allow the other surface to "take over". Given the different enzymatic activities observed for PRC2^CXC>A^ on mono- vs. di-nucleosomes, it would be relevant to know how well this mutant is binding di-nucleosomes. Alternatively, that the authors discuss this aspect in the manuscript.

7) Overall, one limitation in this experimental approach, is that we cannot adequately distinguish between the effect of mutating the specific sites vs. how the mutations may confer overall conformational changes in PRC2 contributing to the altered binding strength. Furthermore, the concluding statement does not take into consideration that the introduction of mutations in the CXC domain may well alter the conformation of EZH2 (or core PRC2) in a way that directly affects the catalytic activity – so loss of H3K27me3 in Figure 2C might also stem from an overall reduction in enzymatic activity rather than (only) from reduced nucleosome binding through this specific surface, especially given the rather modest impact on binding strength shown in Figure 2A. To distinguish between effects on overall catalytic activity and reduced binding, the authors could perform in vitro methyltransferase assays on peptide substrates (as in Figure 3D), using the CXC domain mutant.

8) The interpretations of Figures 4A vs 4B are difficult to follow from the presented data: the Western blot presented in Figure 4B does not strongly support the conclusion of a 3-4-fold global reduction of H3K36me2/me3 (the bands are quite “washed out” in lanes 5-8). The authors further state that the reduction in H3K36me2/me3 is less pronounced in H3K36A embryos vs. H3K36R larvae – and that this might represent methylated H3.3 (which would presumably also be the case for H3K36R), but also maternally-deposited wildtype canonical H3. To substantiate this speculation, the authors should compare the different genotypes in the same developmental stages – thus, a meaningful comparison should include H3K36R mutant embryos (since the H3K36A mutants die before the larval stage).

9) Similarly, it would be interesting to see the ChIP data performed in H3K36R embryos to better evaluate whether the apparently absent (by WB, Figure 4B) or very modest (by ChIP, Figure 4D) reduction of H3K27me3 in H3K36A mutant embryos represents a mechanistic difference between H3K36A vs. H3K36R or simply a difference in developmental stages.

10) How do the authors explain/speculate about the absent/modest effect on H3K27me3 in H3K36A mutant embryos despite the severity of the phenotype for these mutants? Here, again, a side-by-side comparison of H3K27me3 levels in H3K36R and H3K36A mutant embryos would be relevant. It would also be relevant to factor in knowledge about the number of cell divisions and maternal histone load in order to estimate if H3K36A is also impacting H3K27 methylation in vivo.

11) The statement in the Figure 4E legend that "For each HOX gene there is a substantially larger proportion of cells in which the gene is decorated with H3K27me3 and repressed by Polycomb" is probably not correct: While the analyzed tissues are surely a mixed population with few cells showing expression of the tested HOX genes (Figure 5), it is unlikely that the larger proportion of genes show normal H3K27me3 – if this were the case, the ChIP data in Figure 4 would not show the large reduction in H3K27me3 at the tested loci. Thus, the data likely reflects that H3K27me3 is reduced in most of the cell types – yet the de-repression is only seen in a few cells – to show this, immunostaining for H3K27me3 should be included in Figure 5 (discussed further below).

12) The authors state that "the most straightforward interpretation" of the observed misexpression of Polycomb target genes in the H3K36R larvae is that it results from reduced H3K27me3 at these loci. While this may well be the case in this study, loss of H3K27me3 can also be secondary to gene activation – a distinction that is overlooked in many studies. In this case, the question is whether loss of H3K27me3 is a direct effect of the H3K36 substitution or secondary to gene expression changes or altered/delayed development. In this context, the authors could draw more strongly on the previous observation (McKay et al., 2015; Meers et al., 2017) that the H3K36R mutant does not show massive deregulation of gene expression (as mentioned in the Discussion), and, most importantly, should emphasize their own observations that global H3K27me3 is reduced (Figure 4A) and H3K27me3 at Polycomb target genes is strongly reduced (Figure 4C) despite the low percentage of cells assayed showing misexpression of these particular genes (Figure 5B-D). Along with the in vitro data presented in Figure 2, this provides stronger evidence for a direct effect of an “intact” H3K36 in preserving normal PRC2 activity.

13) Along the same lines, it would be interesting to include immunostaining for H3K27me3 in Figures 5B-D to learn whether the loss of H3K27me3 is more prominent in the cells showing mis-expression and to substantiate the data from Figure 4 showing global reduction of H3K27me3.

[Editors' note: further revisions were suggested prior to acceptance, as described below.]

Thank you for submitting your article "Structural basis for PRC2 decoding of active histone methylation marks H3K36me2/3" for consideration by *eLife*. Your article has been reviewed by three peer reviewers, including Jerry L Workman as the Reviewing Editor and Reviewer #1, and the evaluation has been overseen by Cynthia Wolberger as the Senior Editor.

The reviewers have discussed the reviews with one another and the Reviewing Editor has drafted this decision to help you prepare a revised submission.

The authors' revisions have addressed the majority of the comments on the original manuscript, and the additional discussion has clarified their model for how H3K36 interactions with PRC2 may influence its activity. However, there is a remaining point that should be addressed:

The authors' assertion that *Drosophila* Pcl does not bind H3K36me2 or H3K36me3 is not entirely correct. Although the Tudor domain of Pcl does not recognize H3K36me2/3, Ballare et al. demonstrated that a Pcl Tudor-PHD1 construct does bind to H3K36me2 and H3K36me3 (Ballare et al., 2012). This finding is also explicitly stated in their article: "Nevertheless, using a Pcl Tudor-PHD1 construct, we were able to rescue the binding to H3K36me2 and H3K36me3 (Figure 2B)." Furthermore, Musselman et al., 2012, investigated the interactions between the human PHF1 Tudor domain and H3K36me3 but did not test the binding of the *Drosophila* Pcl Tudor-PHD1 domains to H3K36me2 or H3K36me3 (Musselman et al., 2012). In sum, the authors should amend their manuscript to acknowledge that the *Drosophila* Pcl Tudor-PHD1 domains have been shown to bind to H3K36me2 and H3K36me3.

---

## [Author Response]

Essential revisions:1) One of the primary conclusions reached by the authors is that the cleft which H3K36 binds at the EZH2-nucleosomal DNA interface does not fit H3K36A and H3K36R mutations or methylated forms of H3K36 (i.e., H3K36me2 and H3K36me3), leading to allosteric inhibition of PRC2. The structure of the nucleosome-PRC2 complex (Figure 1—figure supplement 4E) illustrates that the side chain of K36 engages in few interactions in this cleft and does not contact the nucleosomal DNA. Furthermore, there is a relatively large unoccupied space in the cleft between the K36 side chain and the DNA, suggesting that amino acids larger than lysine can fit within it. Modeling of an K36R mutation indicates that the arginine side chain can be accommodated in the K36 binding cleft without clashes. Given their similar size, methylated forms of K36 could also fit within the H3K36 binding cleft. The claim that the cleft at the EZH2-DNA interface fits lysine (K36) but cannot accommodate larger amino acids is unconvincing in the absence of additional data.2) The mechanism by which H3K36 binding at the EZH2-nucleosomal DNA interface allosterically regulates PRC2 enzymatic activity is not clearly established. The nucleosome-PRC2 structure shows one potential polar interaction between the K36 side chain and the backbone of Gln570 in EZH2, implying a lack of specificity in recognizing K36. In contrast, other proteins that specifically recognize unmodified lysines in the histone H3 N-terminal tail have been shown to do so through a network of hydrogen bonds and/or salt bridge interactions (e.g., H3K4 recognition by the BHC80 PHD domain; Lan et al. (2007) Nature 448, 718-22). It is unlikely that a single (potential) polar interaction allosterically regulates PRC2 methyltransferase activity.

We agree with the comment that the side chain of an arginine or a tri-methylated lysine or an alanine could theoretically all be accommodated in the cleft between the EZH2^CXC^ domain and the DNA. Our wording in parts of the text that we used in the original manuscript (i.e. the implication that bulkier side chains inhibit) was not well chosen and we replaced the first sentence in the section “Unmodified H3K36 in the EZH2_CXC_-DNA interaction interface is critical for H3K27 methylation in nucleosomes“ by writing “The placement of H3K36 in the EZH2_CXC_-DNA interface (Figure 1F) suggested that even though a tri- or di-methylated K36 side chain could theoretically be accommodated in this interface, these modified side chains might provide a less optimal fit and thereby inhibit H3K27 methylation.”

The reviewer wrote: “K36 engages in few interactions in this cleft and does not contact the nucleosomal DNA.” We are certainly aware that electrostatic interactions between amino acids in proteins are usually considered to require distances of less than 4Å. However, we would like to be cautious and would prefer not to exclude possible (long-range) electrostatic interactions of the K36 side chain with the phosphate backbone of the nucleosomal DNA. First, it should be kept in mind that the global resolution of our structure is in the range of 4.4 Å. Second, what may appear as a “static geometry” in the structure, is certainly not static in solution and the binding geometry may well change the angle of the EZH2 DNA interaction slightly, depending on length of the nucleosomal linker DNA, the sequence of the DNA around the histone octamer etc. We changed the sentence in the section where we describe the structure to include the word “possibly”, i.e. it now reads: “The side chain density of H3K36 suggests that the epsilon-amino group of H3K36 engages in a polar interaction with the carbonyl group of Q570 and possibly in long-range electrostatic interactions with the phosphate backbone of the nucleosomal DNA (Figure 1F, Figure 1—figure supplement 4 C-E).”

It should be emphasized that it is DNA binding that is the driving force for the interaction of PRC2 with nucleosomes (Choi et al., 2017; Wang et al., 2017). The contribution of histone tail interactions to the nucleosome-binding is not really measurable because it is in a completely different range of binding affinity. The Kd of PRC2 for binding to nucleosomes is about 60 nM. The Kd of PRC2 for binding to an unmodified H3[21-41] tail peptide is 3 orders of magnitude higher (29 µM) and for binding to the matching H3K36me3 peptide it is 18 µM (Jani et al., 2019).

We found that PRC2 binds unmodified and H3K36me3-modified nucleosomes with comparable affinity (Figure 3) but that H3K27 methylation on H3Kc36me3 nucleosomes is about 10- to 20-fold less effective than on unmodified nucleosomes. In other words, K27 on K36me3-modified H3 must still be able to engage in a productive manner with the catalytic center, but it is possibly the frequency of productive interactions that is 10- to 20-fold lower. The same is of course true for H3K36R or H3K36A where the inhibition of H3K27me3 formation is about 5-fold (i.e. as shown in Figure 3). So while we agree that H3K36me3, H3K36R or H3K36A can likely be accommodated in the same cleft like H3K36, we however imagine that the residence time of this interaction is diminished and, hence, H3K27 engages with the active site at lower frequency.

In summary, we think that a possible scenario could be that the presence of an unmodified native lysine side chain at position 36 of histone H3 is important within the dynamics of the reaction cycle which involves (i) docking of the complex on the nucleosome via EZH2^CXC^-DNA contacts, (ii) alignment of the H3 N-terminus on the EZH2 surface (iii) docking of the H3K27 side into the catalytic center for methylation of its side chain. This is what we proposed in the Discussion section: “Taken together, a possible scenario would therefore be that within the time frame of the PRC2-nucleosome binding and reaction cycle, docking of the H3K36 side chain in the EZH2_CXC_-DNA interface is critical for rapid alignment of the H3 N-terminus on the EZH2 surface into a catalytically competent state.”

3) The authors attribute the loss of H3K27me3 in the *Drosophila* H3K36A and H3K36R mutants to allosteric inhibition of PRC2. However, these mutations also diminish H3K36me3 in developing *Drosophila* embryos and larvae. The *Drosophila* PRC2 accessory subunit Pcl has been shown to bind H3K36me3, analogous to its mammalian homolog PHF1 (Ballare et al., 2012; Musselman et al., 2012). Thus, the global loss of H3K36me3 may result in either diminished recruitment or mislocalization of PRC2 in chromatin. The authors should examine whether the H3K36A and H3K36R mutations affect PRC2 localization within chromatin relative to wild type *Drosophila*.

The statement “The *Drosophila* PRC2 accessory subunit Pcl has been shown to bind H3K36me3, analogous to its mammalan homolog PHF1 (Ballare et al., 2012; Musselman et al., 2012).” is incorrect. The structure of the tudor domain of *Drosophila* Pcl was determined in 2010 and revealed that Pcl tudor lacks an aromatic cage and does not bind to methylated lysines or arginines (Friberg et al., 2010). The Musselman et al. and Ballare et al. studies in 2012 showed that while the PHF1 tudor domain does bind to H3K36me3 peptides, the tudor domain of *Drosophila* Pcl does not bind H3K36me3 (Ballare et al., 2012, Figure 2B) and that mutation of two aromatic residues in the PHF1 tudor domain that are absent in the Pcl tudor domain abolishes H3K36me3-binding by the PHF1 tudor domain (Musselman et al., 2012).

To remind the reader out this difference, we have added a sentence at the end of the paragraph where we describe the results of HMTase assays with PHF1-PRC2 on H3Kc36me3 nucleosomes:

“In this context it should also be noted that in Polycomblike, the *Drosophila* ortholog of PHF1, the tudor domain contains an incomplete aromatic cage and is unable to bind methylated lysines (Friberg et al., 2010; Musselman et al., 2012; Ballare et al., 2012).”

It should also be kept in mind that, in *Drosophila*, PRC2 is recruited to Polycomb Response Elements (PREs). PRC2 targeting to PREs requires Pho (Wang et al., Mol Cell 2004; Frey et al., G&D 2006) and Pcl (Nekrasov et al., 2007; Savla et al., Development 2008). Moreover, PREs are depleted of nucleosomes (Papp and Müller, G&D 2006; Schwartz et al. JBC 2006 and many more studies since). So there is little foundation for a scenario where H3K36me2/3 would be involved in targeting PCl^-^PRC2.

The reviewer requested to generate PRC2 binding profiles in *Drosophila* with H3K36R or H3K36A chromatin. With the recent worsening of the pandemic situation in Germany, we have not been able to collect the large amount of imaginal disc material that would be needed for perform ChIP-seq analyses for comparing genome-wide binding profiles of PRC2 subunit in wild-type and H3K36R mutant larvae. It is unlikely that we will be able to get that experiment done anytime soon. Considering this and considering that we are in a competitive situation with a manuscript from Eva Nogales’ lab that also reports the H3 binding mode and the role of H3K36, we would ask the reviewers not to insist on including this experiment.

4)The authors use full length human PRC2 in a complex with PHF1 for the Cryo-EM studies and in nucleosome binding experiments. In the Cryo-EM studies, they are not able to detect the density for the N-terminal part of SUZ12, RBBP4 and the N-terminal part of PHF1. In several publications these parts of the PRC2 complex have been shown to be important for the localization of PRC2 to specific sites in the genome. Although, the authors do not test this directly, the N-terminal part of SUZ12 and RBBP4 are most likely important for the binding (affinities) that the authors measure in EMSA. If this indeed is correct, the authors should take this into their discussion of the relative contribution of the various parts of PRC2 to DNA and nucleosome binding affinities.

This is a helpful comment. We extended our discussion of this issue. The text now reads:

“Nucleosome binding by the PRC2^CXC>A/EED>A^ complex could in part be due to incomplete disruption of the mutated interfaces but it likely also represents nucleosome binding contributed by the bottom lobe of PRC2 comprising the N-term of SUZ12 and RBBP4 (Chen et al., 2018; Nekrasov et al., 2005). […] In conclusion, the structural data (Figure 1C, D and Poepsel et al., 2018) suggest that a key interaction of PRC2 with substrate nucleosomes occurs through contacts of the EZH2_CXC_ domain with the DNA gyres, whereas the biochemical analyses argue that this interaction contributes only modestly to the overall chromatin-binding affinity of the complex.”

5) The description in the text states that PRC2^CXC>A^ and PRC2^EED>A^ have similar binding strength deficits in the EMSA assays, which is also shown in the quantification (Figure 2B) – however, based on the gels presented (Figure S5A), the EED mutant appears to perform similarly to the PRC2 WT? Either it is a poorly chosen example (of the three replicates used for quantification) or we would question whether the gel image is of sufficient quality for this quantification.

We agree that the gel in Figure S5A comparing nucleosome binding by PRC2^EED>A^ and the wild-type PRC2 control next to it was not the nicest gel but it was the only gel where we had run the wild-type control directly next to the mutant.

We have removed Figure S5A and now show the EMSA with PRC2^EED>A^ in Figure 2A. The gel image of the PRC2^EED>A^ EMSA (lanes 31-40) has the same experimental setup as in the left panel (lanes 1-30) but was run on a separate gel.

6) The EMSA experiments (Figure 2A and S5A) are performed solely on mononucleosomes, presumably meaning that only one of the studied PRC2 binding surfaces (CXC vs EED) can be engaged at a time. In this scenario, mutation of one binding surface presumably would allow the other surface to "take over". Given the different enzymatic activities observed for PRC2^CXC>A^ on mono- vs. di-nucleosomes, it would be relevant to know how well this mutant is binding di-nucleosomes. Alternatively, that the authors discuss this aspect in the manuscript.

We had performed EMSAs with the aim to analyze binding of PRC2 to dinucleosomes. However, we observed a complex mixture of slowly migrating species and this has precluded further experiments aimed at discriminating between binding events involving specific PRC2 surfaces using dinucleosomes as binding substrates.

We discuss this aspect more extensively in the revised version of the manuscript:

Nucleosome binding by the PRC2^CXC>A/EED>A^ complex could in part be due to incomplete disruption of the mutated interfaces but it likely also represents nucleosome binding contributed by the bottom lobe of PRC2 comprising the N-term of SUZ12 and RBBP4 (Chen et al., 2018; Nekrasov et al., 2005). In particular, biochemical studies on *Drosophila* PRC2 originally found that a minimal complex formed between Su(z)12 and the RBBP4 ortholog Caf1-55 binds to nucleosomes (Nekrasov et al., 2005). Moreover, negative stain EM analyses of human PRC2 bound to a dinucleosome identified several 2D classes where the bottom lobe contacts one or two nucleosomes of the dinucleosome (Poepsel et al., 2018). The binding affinity of the PRC2 core complex for chromatin therefore appears to result from interactions of at least three distinct complex surfaces with nucleosomes, the EZH2_CXC_ domain, the EED nucleosome-binding interface and the SUZ12_N_:RBBP4 lobe. Considering the architecture of the catalytic lobe (Figure 1C) and of the isolated full PRC2 core complex (Kasinath et al., 2018), it is very unlikely that the EZH2_CXC_ domain and the EED nucleosome-binding interface could simultaneously engage with the same nucleosome at a time. Finally, we note that in EMSAs monitoring binding of PRC2 to a dinucleosome, we observed a complex mixture of slowly migrating species and this has precluded experiments aimed at discriminating between binding events involving specific PRC2 surfaces with dinucleosomes. In conclusion, the structural data (Figure 1C, D and Poepsel et al., 2018) suggest that a key interaction of PRC2 with substrate nucleosomes occurs through contacts of the EZH2_CXC_ domain with the DNA gyres, whereas the biochemical analyses argue that this interaction contributes only modestly to the overall chromatin-binding affinity of the complex.

7) Overall, one limitation in this experimental approach, is that we cannot adequately distinguish between the effect of mutating the specific sites vs. how the mutations may confer overall conformational changes in PRC2 contributing to the altered binding strength. Furthermore, the concluding statement does not take into consideration that the introduction of mutations in the CXC domain may well alter the conformation of EZH2 (or core PRC2) in a way that directly affects the catalytic activity – so loss of H3K27me3 in Figure 2C might also stem from an overall reduction in enzymatic activity rather than (only) from reduced nucleosome binding through this specific surface, especially given the rather modest impact on binding strength shown in Figure 2A. To distinguish between effects on overall catalytic activity and reduced binding, the authors could perform in vitro methyltransferase assays on peptide substrates (as in Figure 3D), using the CXC domain mutant.

This is a good control and we have now added this experiment in the revised version. PRC2^CXC>A^ did not show reduced K27 methylation activity compared to wild-type PRC2 on the H3 peptide substrate. The new data are shown in Figure 2—figure supplement 1B and the results are discussed in the text:

“To complement these experiments, we also compared the HMTase activity of wild-type PRC2 and PRC2^CXC>A^ complex on free histone H3_18-42_ peptides using a masspectrometry-based assay to detect H3K27 methylation. It is well known that wild-type PRC2 methylates K27 on free histone H3 with much lower efficiency than on H3 in nucleosomes (Cao et al., 2002; Czermin et al., 2002; Kuzmichev et al., 2002; Muller et al., 2002), and this is also recapitulated in our assays on H3_18-42_ peptides where we primarily detect H3K27me1 but no H3K27me3 formation, even after extended incubation of the reaction (see Figure 2—figure supplement 1B and compare with Figure 2C). However, it is important to note that PRC2^CXC>A^ did not show reduced K27 methylation activity compared to wild-type PRC2 on this H3 peptide substrate (Figure 2—figure supplement 1B). The mutations in the EZH2^CXC^ domain therefore do not appear to alter the conformation of EZH2 in a way that would directly interfere with catalysis.”

8) The interpretations of Figures 4A vs 4B are difficult to follow from the presented data: The Western blot presented in Figure 4B does not strongly support the conclusion of a 3-4-fold global reduction of H3K36me2/me3 (the bands are quite “washed out” in lanes 5-8). The authors further state that the reduction in H3K36me2/me3 is less pronounced in H3K36A embryos vs. H3K36R larvae – and that this might represent methylated H3.3 (which would presumably also be the case for H3K36R), but also maternally-deposited wildtype canonical H3. To substantiate this speculation, the authors should compare the different genotypes in the same developmental stages – thus, a meaningful comparison should include H3K36R mutant embryos (since the H3K36A mutants die before the larval stage).9) Similarly, it would be interesting to see the ChIP data performed in H3K36R embryos to better evaluate whether the apparently absent (by WB, Figure 4B) or very modest (by ChIP, Figure 4D) reduction of H3K27me3 in H3K36A mutant embryos represents a mechanistic difference between H3K36A vs. H3K36R or simply a difference in developmental stages.

We originally had actually analyzed bulk levels of H3K36me2/me3 and H3K27me3 in H3K36R mutant embryos in parallel with the analysis in H3K36A mutant embryos (run the same gel) and we had also performed the H3K36me2 and H3K27me3ChIP-seq analyses in H3K36R mutant embryos. See https://www.biorxiv.org/content/10.1101/2020.04.22.054684v1.full.pdf). We had then performed the analyses in H3K36R mutant larvae where the replacement of wt H3 by H3K36R is more complete and – for simplicity, but perhaps not wisely – had removed the data in H3K36R mutant embryos in the submission to *eLife*. We have now added the western blot and ChIP-seq analysis in H3K36R mutant embryos in Figure 5B, D and F.

Regarding the comment that the “*The Western blot presented in Figure 4B does not strongly support the conclusion of a 3-4-fold global reduction of H3K36me2/me3 (the bands are quite “washed out” in lanes 5-8)*”, we added Author response image 1 with overexposed uncropped images of this western. As illustrated in that figure, there indeed is a clear reduction of H3K36me2 and -me3 in extracts from H3K36A mutant embryos, the western against H2B of the same membrane shown in Author response image 1 serves as control. We estimate the reduction in H3K36me2 and -me3 levels in H3K36A mutant embryos (Figure 4B, lanes 5-8) compared to wild-type embryos (Figure 4B, lanes 1-4) to be about 3-4-fold.

**Author response image 1. sa2fig1:** 

Unexpectedly, we found that in H3K36R mutant embryos the reduction of H3K36me3 is less severe and in the case of H3K36me2 we could not detect a clear reduction (Figure 5B, lanes 9-12). We used the H3K36R strain that we had generated in this study (4 copies of the *3xHisGU^H3K36R^* miniarray) and the effects are therefore directly comparable to the effects observed in H3K36A mutant embryos that carried 4 copies of the *3xHisGU^H3K36A^* miniarray. The less severe reduction in H3K36R mutant embryos compared to H3K36A mutant embryos is also paralleled in the genome-wide H3K36me2 profiling data in these embryos (Figure 5D, violin plots on the left). A simple technical reason such as cross-reactivity of the H3K36me2 and H3K36me3 antibodies specifically with H3K36R seems unlikely given the strong reduction of the H3K36me2 and H3K36me3 signals in extracts from H3K36R mutant larvae (shown in Figure 4A). We currently have no explanation why H3K36me2 and H3K36me3 levels appear to be only very slightly diminished in H3K36R mutant embryos and state this in the text. However, in the ChIP-seq analyses, we nevertheless observe that H3K27me3 levels at HOX are reduced in H3K36R mutant embryos (Figure 4F). Hence, at the HOX genes, H3K27me3 deposition appears to be similarly impaired in H3K36A and H3K36R mutant embryos.These results are described and discussed:

“In *H3^K36A^* mutants, H3K36me2 and H3K36me3 bulk levels were clearly reduced compared to *wildtype* (Figure 5B, compare lanes 5-8 with 1-4). In *H3^K36R^* mutants, H3K36me2 and H3K36me3 bulk levels unexpectedly appeared much less severely reduced (Figure 5B, compare lanes 9-12 with 1-4 and 5-8). As discussed above, the residual H3K36me2 and H3K36me3 signal in *H3^K36A^* and *H3^K36R^* mutant embryos might in part represent modified maternally-deposited canonical wild-type H3 and in part the modified H3.3 variants. However, the reason for the differential reduction of H3K36me2 and -me3 levels in *H3^K36A^* and *H3^K36R^* mutant embryos remains unclear. In both genotypes, H3K27me3 bulk levels appeared largely unchanged compared to *wildtype* (Figure 5B, compare lanes 5-8 and 9-12 with 1-4).”

“As expected from the Western blot analyses (Figure 5B), *H3^K36A^* or *H3^K36R^* mutant embryos showed no general reduction in their genome-wide H3K27me3 profiles (Figure 5D). However, in both mutants, H3K27me3 levels were about 1,5-fold reduced across the HOX gene loci (Figure 5D, F).”

10) How do the authors explain/speculate about the absent/modest effect on H3K27me3 in H3K36A mutant embryos despite the severity of the phenotype for these mutants? Here, again, a side-by-side comparison of H3K27me3 levels in H3K36R and H3K36A mutant embryos would be relevant. It would also be relevant to factor in knowledge about the number of cell divisions and maternal histone load in order to estimate if H3K36A is also impacting H3K27 methylation in vivo.

As discussed in our response to comments 8 and 9, for the western blot and ChIP-seq analyses in embryos, we used the H3K36A and H3K36R strains that we had generated in this study, i.e. *Df HisC* homozygotes rescued by 4 copies of the *3xHisGU^H3K36A^* miniarray and by 4 copies of the *3xHisGU^H3K36R^* miniarray, respectively. So the effects on H3K36me2 and -me3 are therefore directly comparable.

We discuss the difference in lethality in the different mutant strains in the main text:

“The *H3^K36R^* mutant animals from the strain constructed in this study (i.e. containing 4 copies of the *3xHisGU^H3K36R^* miniarray) also completed embryogenesis, and their cuticle morphology was also indistinguishable from *wildtype* (Figure 4). However, 98% of individuals arrested development already at the end of embryogenesis and the 2% of mutant animals that hatched from the eggshell arrested development as first instar larvae (Figure 4). *H3^K36A^* mutants, containing 4 copies of a *3xHisGU^H3K36A^* miniarray, also completed embryogenesis and the morphology of their embryonic cuticle also appeared indistinguishable from *wildtype* (Figure 4). 96% of these *H3^K36A^* mutant animals arrested development before hatching from the eggshell and the 4% that hatched died during the first larval instar (Figure 4). As discussed in the Materials and methods section, the difference in the lethality phase of the *H3^K36R^* and *H3^K36A^* mutants generated in this study compared to *H3^K36R^* mutants in the strain from Matera and colleagues is possibly linked to the histone rescue transgene system used.”

The Materials and methods section contains a more extensive discussion of this point:

“Comparison of the lethality of *H3^K36R^* and *H3^K36A^* mutants: The difference in the lethality phase of the *H3^K36R^* mutants generated in this study compared to *H3^K36R^* mutants in the strain from Matera and colleagues was unexpected because in both strains the *Df(2L)HisC* homozygotes carry 12 copies of the *HisGU^H3K36R^* cassette (i.e. four *3x HisGU^H3K36R^* miniarrays in our strain and a single *12x HisGU^H3K36R^* array in the strain from Matera and colleagues). A possible explanation for the poorer survival of *H3^K36R^* mutants in the strain generated here could be that histone transgene expression from the *3xHisGU^H3K36R^* miniarrays is for some reason less effective that in the case of the *12x HisGU^H3K36R^* array. We also note that a recent study reported that among *Df(2L)HisC* homozygotes that carry 20 *HisGU^H3K36A^* copies, about 50% of the mutant animals develop up to the pupal stages (Zhang et al., 2019). Zhang et al. have not analyzed their *H3^K36A^* mutants any further but it is possible that the higher copy number of the *HisGU^H3K36A^* cassette accounts for the better survival compared to the *H3^K36A^* strain generated in this study. Further studies will be needed to explore whether *H3^K36R^* and *H3^K36A^* mutants show comparable phenotypes in larvae.”

In the main text, we also expanded our explanation about the issue of perdurance of maternally-deposited wild-type histone H3 in H3K36R and H3K36A mutants. The relevant section now reads:

“For the interpretation of the following experiments, it is important to keep in mind that *H3^K36R^* and *H3^K36A^* zygotic mutant animals initially also contain a pool of maternally-deposited wild-type canonical H3 molecules that, together with H3^K36R^ and H3^K36A^, become incorporated into chromatin during the pre-blastoderm cleavage cycles, up to and including the S-phase of cell cycle 14. It is only from the S-phase of cell cycle 15 onwards when only transgene-encoded histones then become incorporated into chromatin (Gunesdogan et al., 2010). Although the wild-type H3 molecules in chromatin become diluted during every cell cycle and are eventually fully replaced by mutant H3, they are probably still present in the chromatin of late-stage embryos. The effective replacement of persisting wild-type H3 molecules by mutant H3 greatly varies between embryonic tissues because of the different numbers of cell divisions that take place in the different tissues prior to the end of embryogenesis. For example, whereas epidermal cells undergo only two more divisions after S-phase 14, certain cells in the CNS undergo as many as 12 divisions (Bossing et al., 1996). In diploid tissues from *H3^K36R^* mutant larvae, replacement of wildtype H3 by H3^K36R^ can be expected to be much more complete because of the extensive cell proliferation that occurs in these tissues that, after metamorphosis, will give rise to the structures of the adult body.”

11) The statement in the Figure 4E legend that "For each HOX gene there is a substantially larger proportion of cells in which the gene is decorated with H3K27me3 and repressed by Polycomb" is probably not correct: While the analyzed tissues are surely a mixed population with few cells showing expression of the tested HOX genes (Figure 5), it is unlikely that the larger proportion of genes show normal H3K27me3 – if this were the case, the ChIP data in Figure 4 would not show the large reduction in H3K27me3 at the tested loci. Thus, the data likely reflects that H3K27me3 is reduced in most of the cell types – yet the de-repression is only seen in a few cells – to show this, immunostaining for H3K27me3 should be included in Figure 5 (discussed further below).

We agree and modified the text to remove any allusion about the proportion of cells where HOX genes are inactive and active. The text now reads: “For every HOX gene, the analyzed tissues (CNS, thoracic imaginal discs and eye-antenna discs) represent a mixed population of cells with a fraction of cells in which the gene is inactive, decorated with H3K27me3 and repressed by PcG and fraction of cells in which the gene is transcriptionally active and carrying the H3K36me2 modification.”

We have added a new data figure (Figure 6—figure supplement 1) that documents that the reduction of H3K27me3 immunofluorescence signal in wing imaginal discs from wild-type and H3K36R mutant larvae. As expected, in H3K36R mutant larvae, the reduction of H3K27me3 is quite uniform across the different cells that form the wing disc. As control for the specificity of the H3K27me3 IF signal, we also show an image of a wing disc with clones of H3K27R mutant cells.

12) The authors state that "the most straightforward interpretation" of the observed misexpression of Polycomb target genes in the H3K36R larvae is that it results from reduced H3K27me3 at these loci. While this may well be the case in this study, loss of H3K27me3 can also be secondary to gene activation – a distinction that is overlooked in many studies. In this case, the question is whether loss of H3K27me3 is a direct effect of the H3K36 substitution or secondary to gene expression changes or altered/delayed development. In this context, the authors could draw more strongly on the previous observation (McKay et al., 2015; Meers et al., 2017) that the H3K36R mutant does not show massive deregulation of gene expression (as mentioned in the Discussion), and, most importantly, should emphasize their own observations that global H3K27me3 is reduced (Figure 4A) and H3K27me3 at Polycomb target genes is strongly reduced (Figure 4C) despite the low percentage of cells assayed showing misexpression of these particular genes (Figure 5B-D). Along with the in vitro data presented in Figure 2, this provides stronger evidence for a direct effect of an “intact” H3K36 in preserving normal PRC2 activity.

This is a good point and we modified the main text to discuss the result from the Meers et al., 2017 study: “these transcriptome analyses did not reveal any gross deregulation of HOX or PcG genes”. We also point out that HOX gene misexpression in *H3^K36R^* or *H3^K36A^* mutants is stochastic and not as widespread as in strong PcG mutants. Specifically, we explain that the stochastic misexpression of *Ubx* in individual cells in the wing blade primordium occurs in an “area of this disc where Ubx is most readily de-repressed if PcG function is perturbed (Beuchle et al., 2001)”. Similarly, we explain that “*Antp* is misexpressed in the antenna primordium, the region of the eye-anntenna imaginal discs where Antp is most susceptible to becoming misexpressed if PcG function is compromised.”

13) Along the same lines, it would be interesting to include immunostaining for H3K27me3 in Figures 5B-D to learn whether the loss of H3K27me3 is more prominent in the cells showing mis-expression and to substantiate the data from Figure 4 showing global reduction of H3K27me3.

As mentioned in our response to Comment 11, we have added a new data figure (Figure 6—figure supplement 1) that documents that the reduction of H3K27me3 immunofluorescence signal in wing imaginal discs from wild-type and H3K36R mutant larvae. As expected, in H3K36R mutant larvae, the reduction of H3K27me3 is quite uniform across the different cells that form the wing disc. As control for the specificity of the H3K27me3 IF signal, we also show an image of a wing disc with clones of H3K27R mutant cells.

We also explain in the text that the stochastic misexpression of *Ubx* in individual cells in the wing blade primordium occurs in an “area of this disc where Ubx is most readily de-repressed if PcG function is perturbed (Beuchle et al., 2001)”. Similarly, we explain that “*Antp* is misexpressed in the antenna primordium, the region of the eye-anntenna imaginal discs where Antp is most susceptible to becoming misexpressed if PcG function is compromised.”

In conclusion, the misexpression of HOX genes is not linked to a particularly severe reduction of H3K27me3 in specific sets of cells or parts of tissues but it simply occurs in stochastic manner and most readily occurs in those cells and tissues where HOX genes also first become de-repressed when Polycomb function is removed (i.e. as illustrated in Beuchle et al., 2001).

[Editors' note: further revisions were suggested prior to acceptance, as described below.]

The authors' revisions have addressed the majority of the comments on the original manuscript, and the additional discussion has clarified their model for how H3K36 interactions with PRC2 may influence its activity. However, there is a remaining point that should be addressed:The authors' assertion that *Drosophila* Pcl does not bind H3K36me2 or H3K36me3 is not entirely correct. Although the Tudor domain of Pcl does not recognize H3K36me2/3, Ballare et al. demonstrated that a Pcl Tudor-PHD1 construct does bind to H3K36me2 and H3K36me3 (Ballare et al., 2012). This finding is also explicitly stated in their article: "Nevertheless, using a Pcl Tudor-PHD1 construct, we were able to rescue the binding to H3K36me2 and H3K36me3 (Figure 2B)." Furthermore, Musselman et al., 2012, investigated the interactions between the human PHF1 Tudor domain and H3K36me3 but did not test the binding of the *Drosophila* Pcl Tudor-PHD1 domains to H3K36me2 or H3K36me3 (Musselman et al., 2012). In sum, the authors should amend their manuscript to acknowledge that the *Drosophila* Pcl Tudor-PHD1 domains have been shown to bind to H3K36me2 and H3K36me3.

We have edited the text to address the remaining point that the reviewers had raised on the revised version.

Specifically, we have clarified what is known about Polycomblike and H3K36me2/3 binding in *Drosophila*. The text now reads:

“In Polycomblike, the *Drosophila* ortholog of PHF1, a region comprising the tudor domain and the adjacent PHD finger has been reported to bind H3K36me3, H3K4me3, H3K9me3 and, more weakly, also H3K14me3 and H3K27me3 (Ballaré et al., 2012). We note, however, that the tudor domain of Polycomblike contains an incomplete aromatic cage and, on its own, is unable to bind methylated lysines (Friberg et al., 2010). Further analyses will be needed to assess whether and how interaction PHF1 or Polycomblike with H3K36me3 might change H3K27 methylation by PRC2 on more complex oligonucleosome substrates that contain both H3K36me3-modified and unmodified nucleosomes.”